# SLITRK2 variants associated with neurodevelopmental disorders impair excitatory synaptic function and cognition in mice

Salima El Chehadeh [1,2,3,33], Kyung Ah Han [4,33], Dongwook Kim [4,33], Gyubin Jang [4,33], Somayeh Bakhtiari [5,6], Dongseok Lim [4], Hee Young Kim [4], Jinhu Kim [4], Hyeonho Kim [4], Julia Wynn [7], Wendy K. Chung [7,8], Giuseppina Vitiello [9], Ioana Cutcutache [10], Matthew Page [10], Jozef Gecz [11,12,13], Kelly Harper [11,12], Ah-reum Han [14], Ho Min Kim [14,15], Marja Wessels [16], Allan Bayat [17,18], Alberto Fernández Jaén [19], Angelo Selicorni [20], Silvia Maitz [21], Arjan P. M. de Brouwer [22], Anneke Vulto-van Silfhout [22,23], Martin Armstrong [24], Joseph Symonds [25], Sébastien Küry [26,27], Bertrand Isidor [26,27], Benjamin Cogné [26,27], Mathilde Nizon [26,27], Claire Feger [28], Jean Muller [3,28], Erin Torti [29], Dorothy K. Grange [30], Marjolaine Willems [31], Michael C. Kruer [5,6], Jaewon Ko [4✉], Amélie Piton [2,28,32✉] & Ji Won Um [4✉]

SLITRK2 is a single-pass transmembrane protein expressed at postsynaptic neurons that regulates neurite outgrowth and excitatory synapse maintenance. In the present study, we report on rare variants (one nonsense and six missense variants) in *SLITRK2* on the X chromosome identified by exome sequencing in individuals with neurodevelopmental disorders. Functional studies showed that some variants displayed impaired membrane transport and impaired excitatory synapse-promoting effects. Strikingly, these variations abolished the ability of SLITRK2 wild-type to reduce the levels of the receptor tyrosine kinase TrkB in neurons. Moreover, *Slitrk2* conditional knockout mice exhibited impaired long-term memory and abnormal gait, recapitulating a subset of clinical features of patients with SLITRK2 variants. Furthermore, impaired excitatory synapse maintenance induced by hippocampal CA1-specific cKO of Slitrk2 caused abnormalities in spatial reference memory. Collectively, these data suggest that SLITRK2 is involved in X-linked neurodevelopmental disorders that are caused by perturbation of diverse facets of SLITRK2 function.

A full list of author affiliations appears at the end of the paper.

ntellectual disability (ID) affects about 3% of the general population and is the most common reason for referral to genetic centers. It is estimated that at least half of cases have a monogenic cause, with more than 1000 genes corresponding to an equivalent number of rare diseases currently identified. The most frequently identified genes are responsible for at most 0.5% of cases. All modes of transmission—autosomal dominant, recessive, and X-linked (dominant or recessive)—are observed. Because of the disproportionate number of males among individuals with ID and the possibility of performing genetic studies of X-linked patterns of inheritance in large families, special attention has been paid to identifying the genetic basis of X-linked ID (XLID). Prior to the advent of next-generation sequencing (NGS), major efforts by the EuroMRX consortium, the Sanger Center, and other international groups encompassing thousands of families[1–4], led to the identification of a large number of XLID genes. The generalization of NGS approaches, including exome sequencing and whole-genome sequencing has greatly accelerated the discovery of autosomal and XLID genes and increased the diagnostic yield of ID, resulting in a shortened diagnostic odyssey for families and expanded access to genetic counseling, including prenatal and pre-implantation genetic diagnosis. Thus, there are now about ~130 identified XLID genes, with an additional ~50 potential candidate XLID genes[5–7]. These include *SLITRK2*, a gene encoding a transmembrane protein that is highly expressed in the brain.

Slit- and Trk-like family proteins, composed of six members (Slitrk1-6), are type I transmembrane proteins that are strongly expressed in the central nervous system[8,9]. Intriguingly, Slitrks positively regulate excitatory and inhibitory synapse development in cultured hippocampal neurons in a paralog-dependent manner[10,11]. Extensive analyses have shown that Slitrks interact with distinct members of the LAR-RPTP (leukocyte common antigen-related receptor protein tyrosine phosphatase) family to promote presynaptic assembly[10,12]. In addition, Slitrk5 interacts with receptor tyrosine kinase TrkB in *cis* to regulate their interactions with LAR-RPTPs[13]. Structural analyses have further demonstrated that the N-terminal leucine-rich repeat 1 (LRR1) of Slitrks interacts with splicing variants of LAR-RPTPs that contain an insert at the MeB site[14,15]. Divergent intracellular mechanisms of Slitrk2 and Slitrk3 also appear to be involved in their synaptic functions[16,17]. Furthermore, systematic analyses to investigate the effects of previously reported *SLITRK* missense variants revealed that a subset of *SLITRK* variants associated with neuropsychiatric disorders[18] disrupt the trafficking of Slitrks to the cell surface and impair their synapse-promoting function[19]. Collectively, these prior studies have established Slitrks as important synapse organizers.

The precise role of the Slitrk2 protein has not yet been clearly defined. The human *SLITRK2* gene is located on the X chromosome, and two missense variants, p. S549F and p.V89M, were reported to be associated with schizophrenia in two female patients who inherited the missense variants from their asymptomatic heterozygous mothers[18]. Furthermore, SLITRK2 interacts with the PDZ domain-containing excitatory scaffold Shank3[17], a gene associated with autism spectrum disorders (ASDs)[20], suggesting its potential involvement in the pathogenesis of neurodevelopmental disorders (NDD).

Here, we report that eight individuals from seven unrelated families with moderate to severe ID and/or NDDs, including neuropsychiatric and behavioral problems, harbor rare potential disease-causing variants of the X-linked gene, *SLITRK2*. We systemically investigated the effects of *SLITRK2* variants on biochemical properties, surface transport, ligand-binding activity, and synaptogenic activities in cultured neurons. In addition, we analyzed the synaptic and behavioral effects of a Slitrk2 deficiency using *Slitrk2* conditional knockout (cKO) mice.

## Results

### Identification of individuals with rare variants in *SLITRK2*.
Exome sequencing, used as a diagnostic approach in individual P1, identified a nonsense c.1381G>T p.E461* variant in the *SLITRK2* gene (Table 1 and Fig. S1a). Sanger sequencing confirmed that this variant was present in his affected brother (P2), but absent from mother's blood, suggesting a de novo occurrence in the maternal germ line (Fig. 1 and Fig. S1a). Because the *SLITRK2* gene contains only one coding exon, we suspected that the mutant transcript does not undergo nonsense mRNA-mediated decay. Through an international data-sharing arrangement initiated through Genematcher[21], we identified six additional individuals with NDDs carrying rare nonsynonymous *SLITRK2* variants predicted to be damaging. Among those were six missense variants—c.221T>C p.L74S (P7), c.628G>A p.E210K (P10), c.934A>G p.T312A (P3), c.1121C>G p.P374R (P8), c.1276C>T p.R426C (P4) and c.1531G>A p.V511M (P5), all found in male individuals (except p.R426C)—that were not previously reported in hemizygous state in gnomAD (Table S1). In all but one of these individuals, no other diagnostic candidate variants were identified during exome sequencing analyses. In P7, a likely pathogenic variant in the homeobox transcription factor *ARX* (c.1109C>T p.A370V) was identified but this did not seem likely to fully explain the patient's phenotype, suggesting that the p.L74S variant identified in *SLITRK2* contributes to the observed symptoms. One variant (p.R426C) occurred de novo, and extended segregation analyses showed that one maternally inherited variant (p.P374R) occurred de novo in the unaffected mother. The remaining variants were maternally inherited. All of these missense variants were predicted to be damaging, based on analyses using SIFT and Polyphen2 (PPH2), with CADD scores between 24.1 and 28.2 (Table S1). The amino acid residues affected by the missense variants are all highly conserved across species and are located in the extracellular part of the SLITRK2 protein (Fig. 1 and Fig. S2).

Seven additional missense *SLITRK2* variants were identified in seven unrelated male patients with NDDs. Among them: (i) a rare missense variant (p.E555D) was identified in a male (P14) who also carried a pathogenic variant in *CUL4B*, encoding a ubiquitin E3 ligase subunit. This *CUL4B* variant appeared sufficient to explain the majority of his clinical manifestations and the *SLITRK2* p.E555D variant was therefore not considered as potentially disease-causing; (ii) four other missense variants—c.601G>A p.V201I (P13), c.931C>G p.P311A (P11), c.1451G>A p.R484Q (P12), and c.2374C>T p.R792C (P9)—were also reported in the hemizygous state in gnomAD (Table S1 and Fig. S1b). They were not therefore likely to be disease-causing, particularly p. V201I variant, which was detected in 19 hemizygous males from gnomAD; and (iii) two additional missense variants were reported in the Decipher database (https://www.deciphergenomics.org/)—c.26G>T, p.S9I and c.44G>A, p.G15E—in individuals with nervous system abnormalities.

### Clinical manifestations in individuals with rare variants in *SLITRK2*.
We collected clinical information for the eight individuals with likely pathogenic variants, P1-5, P7-8, and P10 (Table 1 and Supplementary information). The mean age at examination was 16.9 years (range: 8–30 years). All but one patient had mild to severe intellectual disability and speech delay (mostly severe). Neurological regression was observed in one patient after the age of 6. Patients mainly presented with neuropsychiatric and behavioral issues, including major anxiety (7/8), ASD (4/8), and aggressiveness (2/8). Two patients had attention-deficit hyperactivity disorder (ADHD) with executive impairment, and two were diagnosed with hyperactivity or agitation.

**Table 1 Clinical manifestations identified in individuals harboring potential disease-causing *SLITKR2* variants.**

| | P1 | P2 | P3 | P4 | P7 | P8 | P5 | P10 |
|---|---|---|---|---|---|---|---|---|
| SLITRK2 variant | p.E461* | p.E461* | p.T312A | p.R426C | p.L74S (+*ARX* variant) | p.P374R | p.V511M | p.E210K |
| Sex, age | M, 30 yrs | M, 28 yrs | M, 21 yrs | F, 13 yrs | M 12 yrs | M, 11 yrs | M, 12 yrs | M, 8 yrs |
| OFC (SDS) | >+2 SDS | >+2 SDS | N.A. | <−2 SDS | +1.5 SDS | NA | N.A. | +0.8 SDS |
| Height (SDS) | 0 SDS | 0 SDS | <−5 SDS | −2.5 SDS | +1 SDS | +0.5 SDS | Short stature <−2 DS | +0.5 SDS |
| Feeding difficulties | − | − | + GT | +GOR | − | − | + GOR, PEG | +GOR |
| Kyphoscoliosis | − | − | + | + | + | − | + | − |
| Developmental delay | + | + | Normal before regression | + | + | + | + | + |
| Speech | Delay | Delay | Regression | Absent | Delay | Delay | Delay | Language impairment |
| ID | Mild | Moderate to severe | Severe | Severe | Moderate to severe | Mild | Moderate to severe | Border line |
| Seizure (age at onset) | − | − | + (10 yrs) | + (11 mths) | − | + (8 yrs) | − | − |
| Seizures type | − | − | Multifocal, prominent in left central region | Generalized seizures | − | Focal seizures | − | − |
| Spasticity/Dystonia | − | − | Spasticity | Diplegic cerebral palsy | − | − | Dystonic diplegia | − |
| Unsteady gait | − | − | + a | + c | + | − | + g | − |
| Neuropsychiatric manifestations | Anxiety | Major anxiety, ASD, mutism, aggressiveness | − | ASD, anxiety, mutism, aggressiveness | ASD, anxiety, hyperactivity | Anxiety, ADHD, very sensitive, easily frustrated | Anxiety, ADHD | ASD, severe anxiety, ADHD, OCD, vocal tics |
| Other Brain MRI anomalies | Normal | Normal | + b | + d | Not done | + f | + e | Not done |

a: Confined to a wheelchair at 21 yrs; b: severe cerebral and cerebellar volume loss, ventricle dilation, atrophy of corpus callosum and brainstem, bilateral hippocampal atrophy with increased FLAIR signal; c: walking with support; d: thin corpus callosum, white matter diffuse reduction, and leukomalacia; e: paucity of white matter, bilateral periventricular lesions, lateral ventricles dilation; f: unspecific minor white substance punctate changes on the right side around the trigonum; g: uses a walker and wheelchair for getting around.
*ADHD* attention-deficit hyperactivity disorder, *ASD* autism spectrum disorder, *GOR* gastroesophageal reflux, *GT* gastric tube feeding, *mths* months, *MRI* magnetic resonance imaging, *OCD* obsessive-compulsive disorder, *OFC* occipitofrontal circumference, *PEG* percutaneous endoscopic gastrostomy, *SDS* standard deviation score, *yrs* years.

One patient had obsessive-compulsive behavior, vocal tics, "meltdowns" and tantrums. We also noted an unsteady gait (4/6), spasticity and/or dystonia (3/8), and seizures (3/8). Several patients (4/8) had kyphoscoliosis. Three patients had short stature (≤2 SDS). The patients for whom we were able to get photographs were not clearly dysmorphic except for one patient (Supplementary information). Occipitofrontal circumference measurements were highly variable, including two patients with macrocephaly and one patient with microcephaly. Brain magnetic resonance imaging (MRI) of most patients was normal or revealed nonspecific anomalies (4/8), including white matter reduction, ventricular dilation, and thin corpus callosum; three patients had white matter anomalies (Table 1).

**Biochemical, structural, and ligand-binding phenotypes of disease-associated *SLITRK2* variants**. To investigate the structural effect of patient variants on human SLITRK2, we predicted the structure of SLITRK2 using AlphaFold2[22,23]. Three variants (L74S, V201I, and E210K) are located in the LRR1 domain (aa C33-D270) and six variants (P374R, R426C, E461*, R484Q, V511M, and E555D) are located in the LRR2 domain (Fig. 2, top). Free energy calculations indicated that three variants (L74S, P374R, and V511M) are predicted to affect the structural stability of the protein (Table S2), possibly causing misfolding and aberrant trafficking. In particular, a point mutation in the nonpolar leucine, L74, to a polar serine residue in the consensus LxxLxLxxNxL motif of LRR1-1 is expected to disrupt hydrophobic interactions with neighboring nonpolar residues (V54, L69, L77, L91, and L93). Similarly, mutation of P374 to arginine is expected to disrupt the LRRNT hydrophobic core of LRR2, comprising with V357 and L380. A point mutation, resulting in the replacement of a small, nonpolar valine by a long methionine (V511M), may collapse LRRCT folding of the LRR2 domain. The crystal structure of the human SLITRK1 LRR1 domain in complex with PTPδ Ig1-3 has shown that E206 on the concave surface of SLITRK1 makes ionic interactions with the side chain of K135 and main chain of V136[14]. Mutation of SLITRK2 E210 corresponding to SLITRK1 E206 to positively charged lysine would be expected cause charge repulsion, thus interrupting the interaction of SLITRK2 LRR1 with LAR-RPTPs. Because cysteine residues are usually undesirable in exposed regions of a protein unless they acquire functions such as metal binding[24], the R426C variant on the concave surface may adversely affect protein stability. However, other variants exposed on the surface (E461*, R484Q, and E555D in LRR2) and V201I, which form hydrophobic networks in LRR1, are predicted to have little effect on the three-dimensional (3D) structure of SLITRK2.

To further investigate the impact of amino acid substitutions on the structure and/or synaptic function of SLITRK2, we generated mammalian expression vectors encoding *SLITRK2* variants identified in individuals with NDD, both those considered potentially disease-causing and those identified in gnomAD (Table S1 and Fig. 3a). As controls, in addition to the

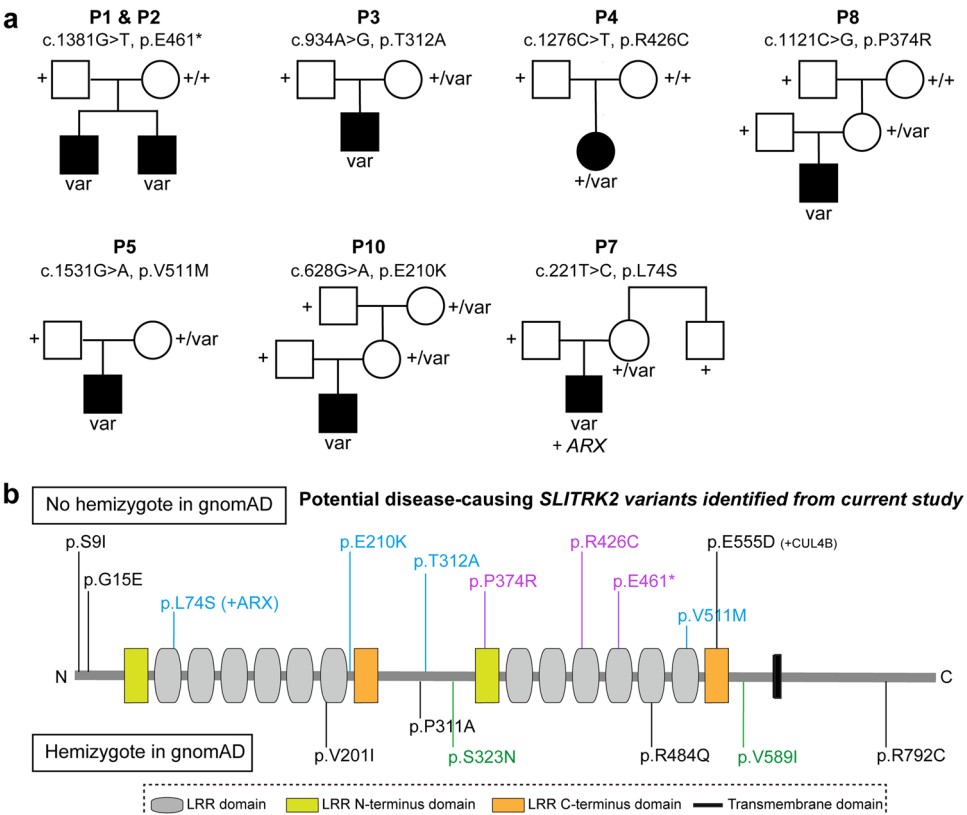

**Fig. 1 Identification of *SLITRK2* variants in individuals with NDD. a** Pedigree of families with rare *SLITRK2* variants. **b** Representation of the SLITRK2 protein, including the different variants identified in individuals with NDD and not previously reported in the hemizygous state in the gnomAD database (Purple, variants occurring de novo in the proband or his mother; blue, variants inherited from unaffected mothers without additional information; green, variants identified in gnomAD in at least two hemizygous males).

V201I variant, we included two other variants—c.967A>G p.S323N and c.1765G>A p.V589I—present in 6 and 139 males from gnomAD respectively and therefore presumed to be benign (Table S1). Notably, P374 and V511 residues in human SLITRK2 are conserved in other human SLITRK proteins at equivalent positions, but none of the other residues exhibit complete sequence identity across the six SLITRK members (Fig. S2a). The two variants from Decipher, S9I and G15E, are located in the signal peptide of SLITRK2, which is expected to be cleaved after membrane targeting. Thus, for these two variants, we generated pCMV5-SLITRK2 vector harboring its own signal peptide, whereas the other variants were generated using the pDisplay-SLITRK2 vector (Fig. 3a). Most of these residues are quite evolutionarily conserved among various vertebrate species, implying their possible functional significance (Fig. S2b).

We then examined the expression levels and intracellular transport properties of the various SLITRK2 variants following expression in human embryonic kidney 293 T (HEK293T) cells (Fig. 3). Immunoblot analyses of lysates of HEK293T cells transfected with vectors encoding HA-tagged full-length SLITRK2 wild-type (WT) or the respective variants showed that total protein expression levels of all SLITRK2 point mutants (except E461*) were comparable to those of SLITRK2 WT (Fig. 3b). The E461* variant, which resulted in a premature termination codon, yielded a truncated protein of ~60 kDa in SDS-PAGE analyses (Fig. 3b). Notably, unlike SLITRK2 WT, which is expressed as both N-glycosylated and non-glycosylated protein species, three SLITRK2 variants (L74S, P374R, and R426C) in HEK293T cells were detected as immature, non-glycosylated forms (Fig. S3). We next examined the surface and

intracellular protein levels of SLITRK2 WT and the indicated SLITRK2 variants in HEK293T cells (Fig. 3c, d). In keeping with the glycosylation profiles (Fig. S3), only three SLITRK2 variants (L74S, P374R, and R426C) displayed significantly reduced surface levels, reflecting their complete entrapment in an intracellular compartment (Fig. 3c, d), in the absence of altered cellular membrane integrity (as monitored by transferrin receptor expression; Fig. S4). The E461* variant also exhibited reduced surface levels (Fig. 3c, d), in agreement with the absence of a transmembrane region in this variant; thus, this truncated SLITRK2 is likely secreted (Fig. S5). Results obtained using surface biotinylation assays were consistent with these findings, showing markedly lower surface expression levels of L74S, P374R, R426C, and E461* variants (Fig. S6). Note that, although S9I and G15E variants are located in the SLITRK2 signal peptide, they did not affect the membrane transport of SLITRK2 (Fig. 3c, d).

To determine whether the various *SLITRK2* variants affected the interactions with PTPδ, an extracellular ligand of SLITRK2, we assayed cell-surface binding of recombinant Ig-fusion proteins of PTPδ (IgC-PTPδ) or IgC alone (negative control) with HEK293T cells expressing HA-tagged SLITRK2 variants (Fig. S7). IgC-PTPδ proteins avidly bound to HEK293T cells expressing SLITRK2 variants, except those expressing a subset of SLITRK2 variants with defective surface transport properties (Fig. S7). IgC did not bind to any tested SLITRK2 variants (Fig. S7).

To ensure that the subset of the SLITRK2 variants with impaired surface transport properties exhibited similar trafficking defects in neurons, we transfected cultured hippocampal neurons with WT or variant SLITRK2 and analyzed them by immunocytochemistry. Whereas most HA-tagged SLITRK2 variants were transported into

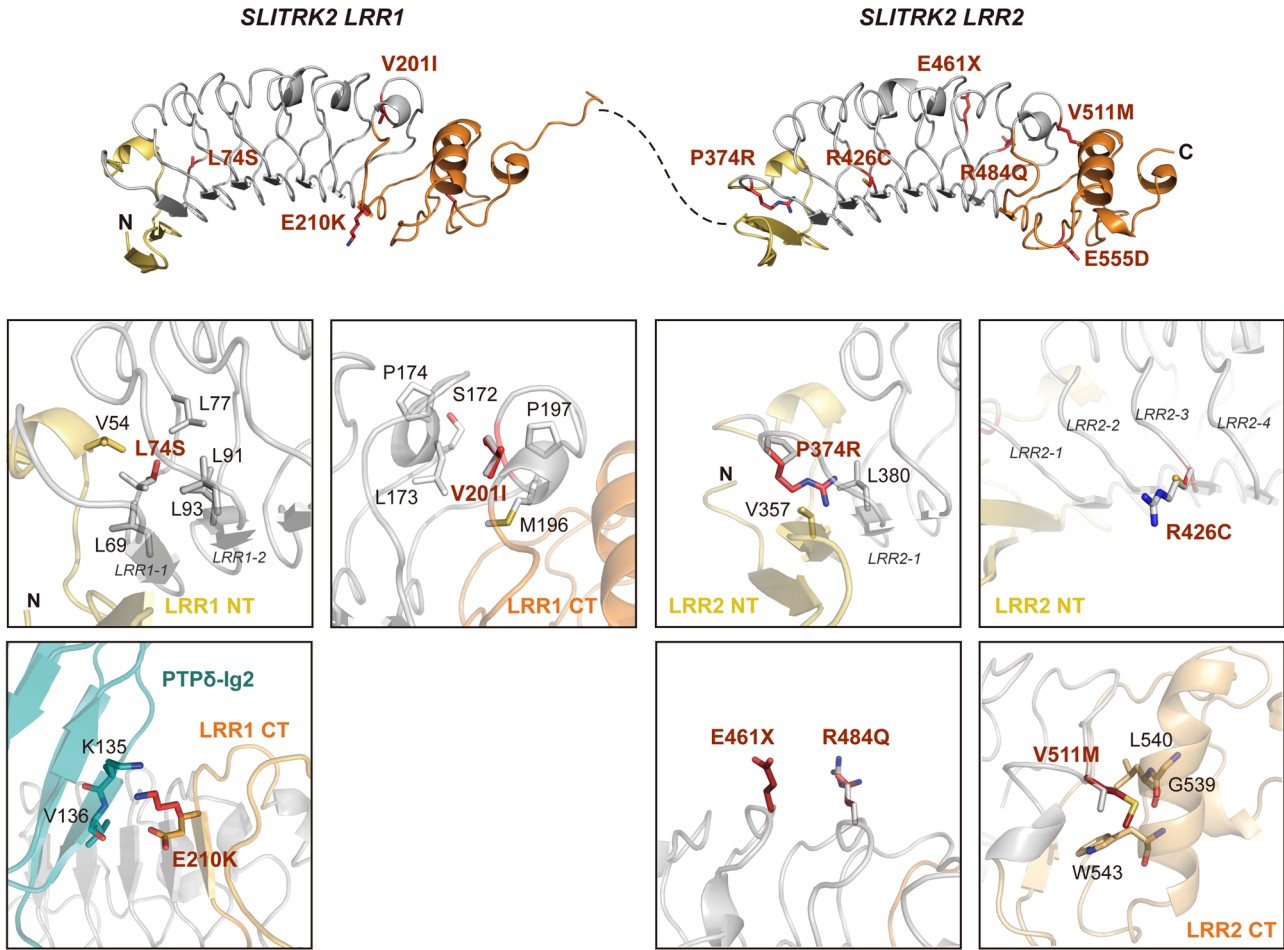

**Fig. 2 Structural model of human SLITRK2 and patient variants.** Model structure of human SLITRK2 LRR1 and LRR2 domains presented as a cartoon. Top: LRR1 domain, C33-D270 (left); LRR2 domain, P341-P579 (right). LRR N-terminus, LRR motifs, and LRR C-terminus are depicted in yellow, gray, and orange, respectively. Black dotted lines represent flexible linkers between LRR1 and LRR2 domains. Bottom: Close-up views of mutated and neighboring residues in LRR1 and LRR2 domains. Residues corresponding to patients' variants described in the current study are presented as red sticks and labeled. The structure of human PTPδ Ig2 (teal) from the crystal structure of the human SLITRK1/human PTPδ Ig1-3 (PDB ID:4RCA) complex was used as a reference for potential interactions between human SLITRK2 and PTPδ Ig2. For clarity, the human SLITRK2 model is rotated 90° around the x-axis in the close-up views of E210K, E461*, R484Q, and V511M.

dendrites as efficiently as SLITRK2 WT (Fig. 4), three SLITRK2 variants (L74S, P374R, and R426C) were either completely retained in the cell body of transfected neurons, or displayed markedly impaired trafficking into dendrites (Fig. 4), as judged relative to immunofluorescence signals in the somatic compartment of transfected neurons. Puzzlingly, HA immunoreactive intensity in the somatic compartment was not decreased in neurons transfected with L74S, P374R, or R426C (see Fig. 4b), possibly reflecting imprecise evaluation owing to saturation of HA immunoreactivity in transfected neurons. We thus repeated the same experiments using a subset of SLITRK2 variants and quantified HA immunoreactive intensity in transfected neurons imaged using settings that prevented saturation of immunofluorescence signals (Fig. S8a, b). We found that HA immunoreactivity was significantly increased in somatic compartments of neurons expressing L74S or P374R, but not in those of neurons expressing R426C (Fig. S8a, b). Similar findings were also obtained using Slitrk2-deficient cultured neurons (Fig. S9). These findings are also consistent with the slight increase in R426C entrapment in an intracellular compartment (i.e., cis-golgi) of cultured neurons, as assessed by its colocalization with anti-GM130 (Fig. S10). No alterations in dendritic branching were observed in neurons expressing SLITRK2 WT or the indicated SLITRK2 variants (Fig. S8c). The E461* variant also exhibited

impaired dendritic trafficking and reduced levels of HA immuno-fluorescence signals in the somatic compartment (Fig. 4), in agreement with the absence of a transmembrane region; in E461*, again suggesting that the truncated SLITRK2 protein is likely secreted. These observations suggest that the surface transport deficiency observed in heterologous cells is similarly recapitulated in cultured hippocampal neurons. Taken together, these data suggest that one subset of SLITRK2 variants observed in patients with neurodevelopmental disorders causes improper biochemical processing and abnormal trafficking.

**Synaptic phenotypes of disease-associated SLITRK2 missense variants.** SLITRK2 was previously shown to trigger presynaptic differentiation when expressed in heterologous cells and cocultured in contact with axons[10,12,19]. To test whether the surface transport-defective SLITRK2 variants exhibit differences in presynaptic differentiation-inducing ability, we performed heterologous synapse-formation assays using HEK293T cells expressing SLITRK2 WT or the indicated SLITRK2 variants (Fig. S11). SLITRK2 WT robustly recruited clustering of both the glutamatergic presynaptic marker VGLUT1 (vesicular glutamate transporter 1) and GABAergic pre-synaptic marker GAD67 (glutamic acid decarboxylase 67 kDa)

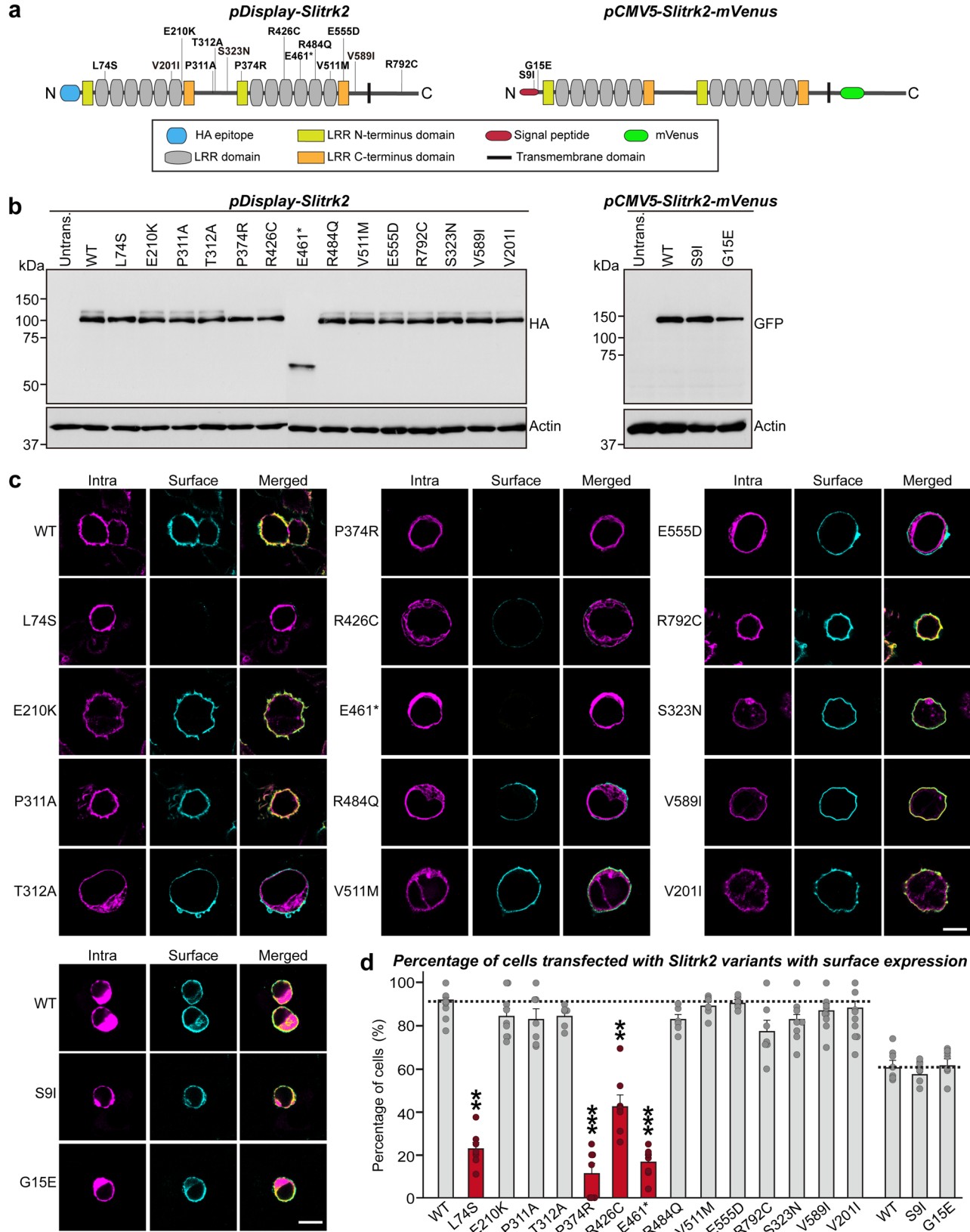

around SLITRK2-expressing HEK293T cells, whereas L74S, P374R, R426C, and E461* were functionally inactive in recruiting both VGLUT1 and GAD67, as expected given their impaired surface expression levels and PTP-binding activities (Fig. S11; see Figs. 3c, d, S6 and S7).

**Generation of *Slitrk2*-cKO mice.** To evaluate the cellular effects of SLITRK2 variants, we generated conditional *Slitrk2*-KO (*Slitrk2*-cKO) mice. Transgenic mice with a deletion of *Slitrk2* were generated by crossing homozygous *Slitrk2*-floxed mice, in which exon 2 and exon 3 are flanked by loxP sites, with a Cre

**Fig. 3 Impaired surface trafficking of a subset of SLITRK2 variants. a** Schematic diagrams depicting the *SLITRK2* variants analyzed in the current study. **b** Representative immunoblots from HEK293T cells transfected with the indicated WT or mutant forms of SLITRK2. Samples containing equal amounts of protein were resolved by SDS-PAGE and immunoblotted using anti-HA or GFP antibodies; β-actin was used as a loading control. Molecular mass markers are labeled in kilodaltons. The experiments were independently repeated three times. **c** Surface expression analysis of HEK293T cells expressing the indicated WT or mutant forms of SLITRK2. Transfected cells were immunostained with mouse anti-HA antibodies (cyan) and detected with FITC-conjugated anti-mouse secondary antibodies under non-permeabilized conditions, followed by permeabilization of cells. Cells were then stained first with rabbit anti-HA antibodies (magenta) and then with Cy3-conjugated anti-rabbit secondary antibodies. Scale bar, 10 μm (applies to all images). **d** Quantification of the proportion of cells exhibiting surface expression. All data are shown as means ± SEMs ('*n*' denotes the number of images from three independent experiments; WT, *n* = 10; L74S, *n* = 7; E210K, *n* = 9; P311A, *n* = 7; T312A, *n* = 6; P374R, *n* = 7; R426C, *n* = 8; E461*, *n* = 8; R484Q, *n* = 6; V511M, *n* = 6; E555D, *n* = 6; R792C, *n* = 7; S323N, *n* = 13; V589I, *n* = 13; V201I, *n* = 16; WT, *n* = 7; S9I, *n* = 7; and G15E, *n* = 7; **$p < 0.01$, ***$p < 0.001$; ANOVA with a non-parametric Kruskal–Wallis test). See Source data for raw data values and Supplementary Table 4 for statistical details.

recombinase driver line under control of the Nestin promoter (*Nestin*-Cre) (Fig. 5a, b). Quantitative RT-PCR analyses showed that mRNA levels of *Slitrk2*, but not those of the other *Slitrks*, were completely eliminated in Nestin-*Slitrk2* mice (Fig. 5c). Nestin-*Slitrk2* male mice weighted marginally (but significantly) less at postnatal day 30 (P30), P40, and P50 compared with control mice, a generalized metabolic phenotype of the Nestin-Cre driver line[25] (Fig. 5d), although their brain weights and size were comparable (Fig. 5e, f). Gross morphology, cortical thickness and neuron numbers, including interneuron numbers, were comparable between Nestin-*Slitrk2* mice and control littermates, as assessed by Nissl staining (Fig. 5g) and immunostaining for the neuron-specific marker NeuN (Fig. 5h, i) and GABAergic interneuron marker GAD67 (Fig. 5j, k). In addition, semi-quantitative immunoblot analyses confirmed complete elimination of detectable Slitrk2 protein in Nestin-*Slitrk2* mice (Fig. 5l, m), as assessed with the Slitrk2-specific polyclonal antibody, JK177 (Fig. S12), and further showed that expression levels of various synaptic proteins in the cortex of Nestin-*Slitrk2* mice were comparable to those of littermate control mice (Fig. 5l, m). Moreover, ventricle volumes in Nestin-*Slitrk2* mice were comparable to those of control mice (Fig. S13).

**Slitrk2-cKO induces an impairment in excitatory synaptic development that is not reversed by reintroduction of a subset of SLITRK2 patient variants.** Slitrk2 was previously shown to specifically promote excitatory, but not inhibitory, synapse maintenance in cultured neurons and hippocampal CA1 neurons[10,17,19]. To address this, we first examined whether Slitrk2 deletion induced a specific decrease in excitatory synapse numbers and synaptic transmission in cultured hippocampal neurons. Immunostaining neurons at DIV14 with antibodies to VGLUT1 and SHANK, markers for excitatory postsynaptic density, revealed a reduction in the density of excitatory synaptic puncta immunoreactive to both VGLUT1 and SHANK (Fig. S14a, b). As expected, neither the density nor size of inhibitory synaptic puncta was changed in neurons immunostained with antibodies to VGAT and GABA$_A$γ2, an inhibitory postsynaptic marker (Fig. S14a, b, d).

To assess whether the potential disease-causing SLITRK2 variants affect the excitatory synapse-promoting function of SLITRK2 WT, we infected cultured hippocampal *Slitrk2*-cKO neurons with lentiviruses expressing either inactive Cre (ΔCre; Control) or active Cre recombinase at DIV5 and co-transfected them with human SLITRK2 expression variants and EGFP at DIV8–9. We subsequently immunostained these infected/co-transfected neurons with antibodies to synaptophysin, SHANK and EGFP (to visualize the transfected neurons) at DIV13–14. The density, but not the size, of excitatory synaptic puncta immunoreactive to both synaptophysin and SHANK was significantly reduced in *Slitrk2*-deficient neurons (Fig. 6a, b; Fig. S14e). Co-expression of SLITRK2 WT completely reversed the excitatory

synaptic deficits observed in *Slitrk2*-deficient neurons (Fig. 6a, b). In contrast, co-expression of the SLITRK2 variants, L74S, P374R, R426C, or E461*, consistently failed to rescue the deficits in excitatory synapses (Fig. 6a, b), in agreement with the results obtained in heterologous synapse-formation assays (Fig. S11). Strikingly, expression of T312A also failed to rescue the deficits in excitatory synapses (Fig. 6a, b), despite the fact that this SLITRK2 variant exhibited expression levels, surface transport, and presynaptic differentiation-inducing abilities comparable to those of Slitrk2 WT (see Figs. 3, 4, S4, S6, and S11).

To corroborate these immunocytochemical results, we measured miniature excitatory postsynaptic currents (mEPSCs) in cultured hippocampal neurons using whole-cell electrophysiological recordings (Fig. 6c–e). We found that the frequency, but not amplitude, decay time or rise slope, of mEPSCs was markedly decreased in *Slitrk2*-deficient cultured neurons (Fig. 6c–g; see also Fig. S15). In contrast, Slitrk2 deletion did not affect the frequency or amplitude of miniature inhibitory postsynaptic currents (mIPSCs) in cultured hippocampal neurons (Fig. S15). Co-expression of SLITRK2 WT or the indicated SLITRK2 variants completely reversed decrease in mEPSC frequency (Fig. 6c, e), whereas co-expression of SLITRK2 variants defective in rescuing anatomical deficits failed to rescue mEPSC frequency (Fig. 6c, e). Consistent with results obtained in cultured hippocampal neurons, immunohistochemical analyses using Nestin-*Slitrk2* mice revealed that the density of VGLUT1 and PSD-95 puncta was significantly decreased in a subset of hippocampal CA1 layers (*stratum oriens* [SO] and *stratum radiatum* [SR]) (Figs. S16). These results collectively suggest that the synaptic phenotypes of the various *SLITRK2* variants likely reflect perturbation of diverse facets of SLITRK2 functions.

**Reduced TrkB levels induced by SLITRK2 WT, but not by disease-associated SLITRK2 variants.** Previous studies have shown that SLITRK5 acts as a co-receptor for TrkB that orchestrates brain-derived neurotrophic factor (BDNF)-TrkB-dependent trafficking and signaling[13], and that an obsessive-compulsive disorder (OCD)-associated *SLITRK5* missense variant impairs SLITRK5 binding to TrkB[26]. Thus, we next analyzed whether the identified *SLITRK2* variants also altered TrkB binding, assessed by coimmunoprecipitation assays in HEK293T cells. We found that disease-linked *SLITRK2* variants with impaired excitatory synapse maintenance did not affect the ability of SLITRK2 to bind TrkB (Fig. S17a, b).

BDNF plays critical roles in neuronal survival and synaptic plasticity through activation of a full-length receptor (TrkB-FL) while truncated TrkB isoforms (TrkB-T) are presumed to competitively inhibit BDNF-mediated TrkB signaling in certain contexts[27]. Excitotoxic stimulation of cultured neurons with glutamate downregulated TrkB-FL, resulting in downregulation of BDNF-TrkB signaling[28]. In addition, calpain activation triggers the truncation of TrkB-FL, leading to downregulation of BDNF-

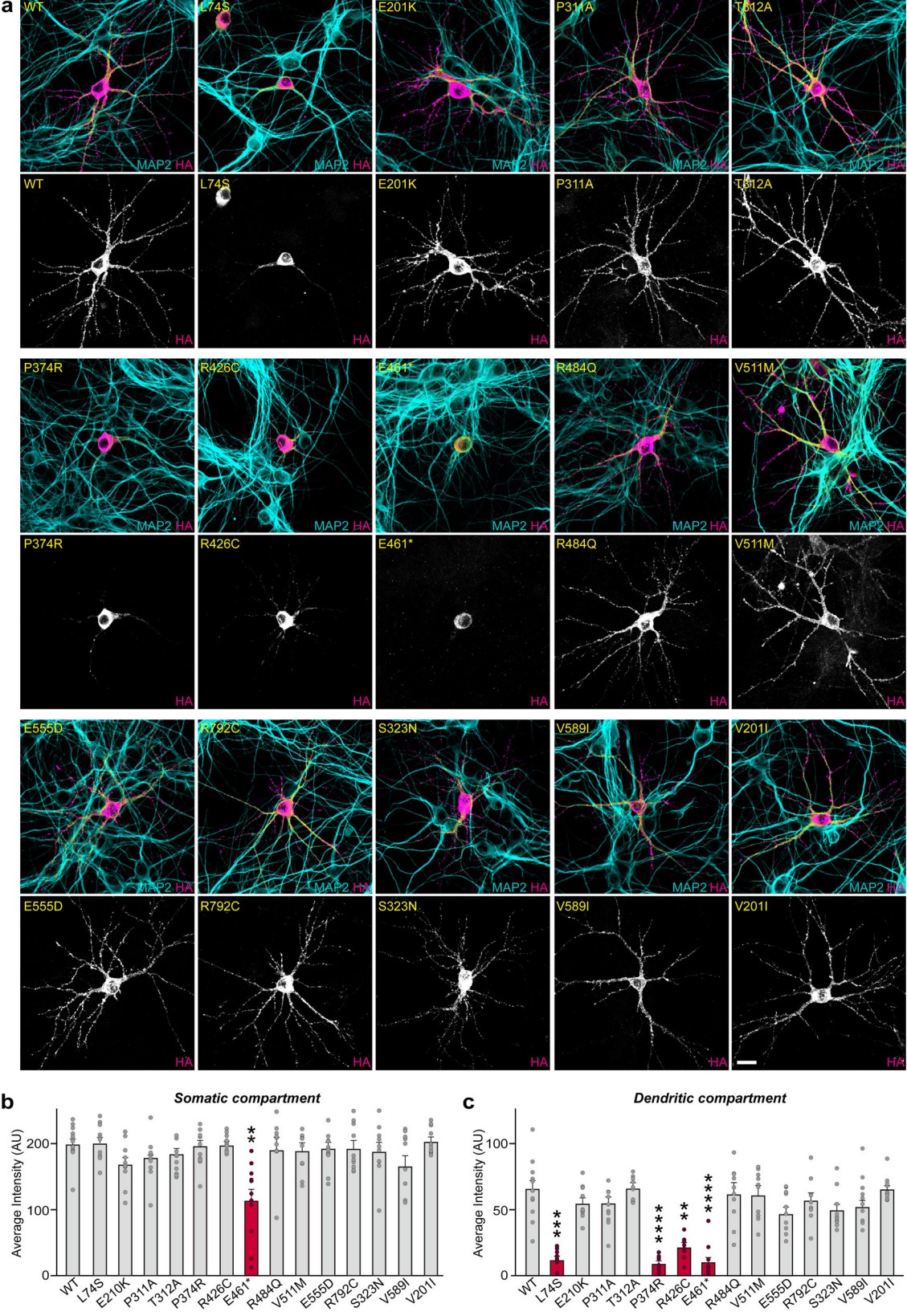

TrkB signaling[29]. These findings suggest that disease-linked *SLITRK2* variants could alter BDNF-TrkB signaling. To test this possibility, we infected DIV3 cortical cultured neurons with lentiviruses expressing SLITRK2 WT or the indicated variants, and collected the infected neuronal lysates at DIV14. Immunoblot analyses revealed that overexpression of SLITRK2 WT

significantly decreased TrkB-FL levels, but markedly increased TrkB-T levels (Fig. 7a, b), suggesting that SLITRK2 might reduce forward trafficking of TrkB and/or trigger the truncation of TrkB-FL. Strikingly, T312A abolished the ability of SLITRK2 WT to decrease TrkB-FL levels (Fig. 7a, b). In addition, the effects of L74S, P374R, R426C, or E461* were similar to those of T312A

**Fig. 4 Impaired dendritic targeting of a subset of SLITRK2 variants in cultured hippocampal neurons. a** Representative images of hippocampal neurons transfected with the indicated WT or mutant forms of SLITRK2 at DIV10. The transfected neurons at DIV14 were double-immunostained for antibodies against the somatodendritic marker, MAP2 (cyan), and HA (magenta). Scale bar, 20 μm (applies to all images). **b, c** Somatic (**b**) or dendritic (**c**) targeting of WT or the indicated mutant forms of SLITRK2 in hippocampal neurons was quantified by measuring average intensity of HA immunofluorescence in primary dendrites. The average intensities of SLITRK2 WT and variants in the soma region of transfected neurons were also quantified. All data are shown as means ± SEMs ('$n$' denotes the number of neurons from three independent experiments; WT, $n = 12$; L74S, $n = 10$; E210K, $n = 10$; P311A, $n = 10$; T312A, $n = 10$; P374R, $n = 11$; R426C, $n = 10$; E461*, $n = 11$; R484Q, $n = 10$; V511M, $n = 10$; E555D, $n = 11$; R792C, $n = 10$; S323N, $n = 10$; V589I, $n = 10$; V201I, $n = 10$; **$p < 0.01$, ***$p < 0.001$, ****$p < 0.0001$; ANOVA with a non-parametric Kruskal–Wallis test). See Source data for raw data values and Supplementary Table 4 for statistical details.

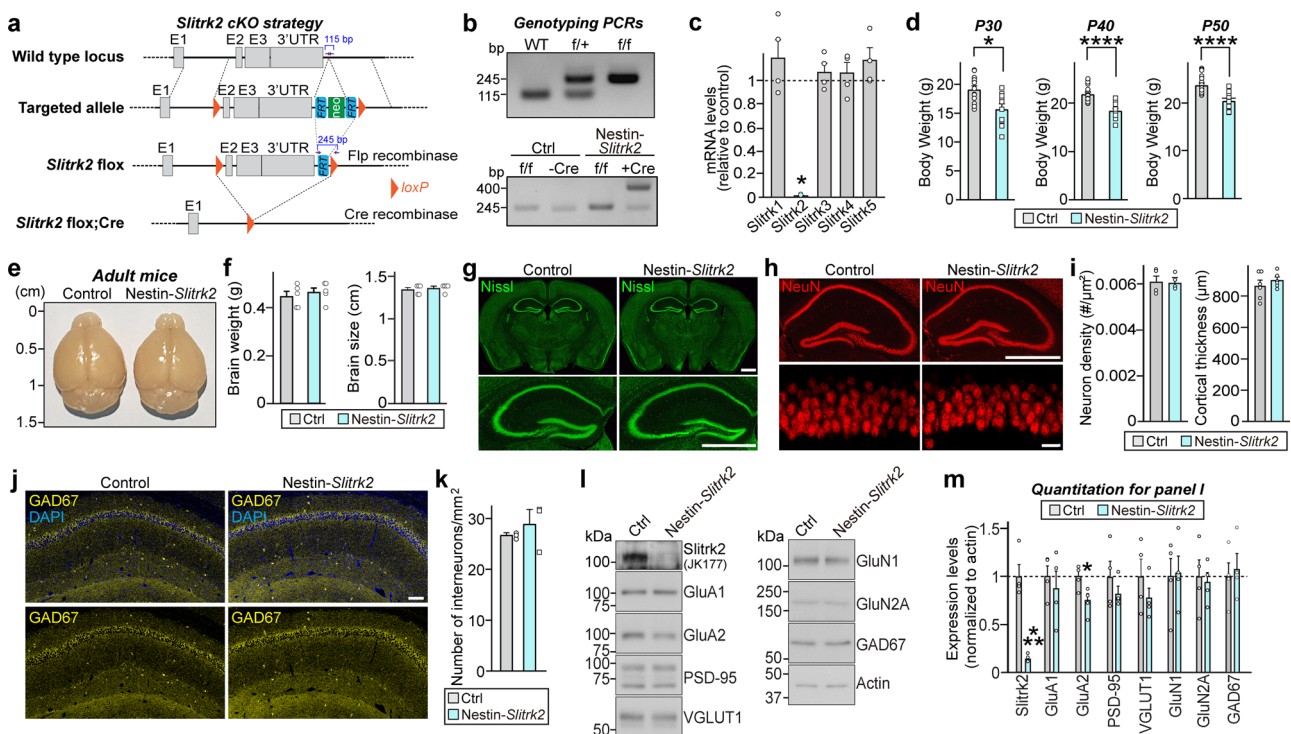

**Fig. 5 Generation of *Slitrk2*-cKO mice. a** Strategy used to generate *Slitrk2*-cKO mice. LoxP sites were introduced at positions flanking the neomycin gene, FLP recombinant target (FRT), and exon 2–3 (E2 and E3) of the *Slitrk2* gene. Black arrows indicate forward and reverse primers used for genotyping. Note that lacZ and neomycin cassettes are two separate markers. **b** PCR genotyping of *Slitrk2*-floxed mice (top) and Nestin-Cre;*Slitrk2*-cKO (Nestin-*Slitrk2*) mice (bottom). The band size for the *Slitrk2*-floxed allele was 245 bp. The 400-bp Cre-specific PCR product was detected only in Nestin-*Slitrk2* mice. **c** Quantitative RT-PCR analyses of *Slitrk* mRNA levels in Nestin-*Slitrk2* mice. Data are mean ± SEMs ($n = 4$ mice; *$p < 0.05$; two-tailed Mann–Whitney $U$ test). **d** Body weights of Nestin-*Slitrk2* and *Slitrk2*-floxed (Ctrl) mice at P30, P40, and P50. Data are means ± SEMs ('$n$' denotes the number of mice; P30: Ctrl, $n = 15$; Nestin-*Slitrk2*, $n = 10$; P40: Ctrl, $n = 20$; Nestin-*Slitrk2*, $n = 14$; P50: Ctrl, $n = 20$; Nestin-*Slitrk2*, $n = 14$; *$p < 0.05$, ****$p < 0.0001$; two-tailed Mann–Whitney $U$ test). **e, f** Representative photographs of whole brains (**e**) and quantification of brain weights and size (**f**) of P42 Nestin-*Slitrk2* and *Slitrk2*-floxed (Ctrl) mice. Data are means ± SEMs ('$n$' denotes the number of mice; Ctrl and Nestin-*Slitrk2*, $n = 5$; two-tailed Mann–Whitney $U$ test). **g** Normal gross morphology of the Nestin-*Slitrk2* brain at 7 wks, as revealed by Nissl staining. Scale bar: 1 mm (applies to both top and bottom). **h** Representative images of NeuN (a neuronal marker) staining in the Nestin-*Slitrk2* brain. Normal numbers of neurons in hippocampal regions. Top, hippocampus; bottom, CA1 *stratum pyramidale* layer. Scale bar: 1 mm (top) and 20 μm (bottom). **i** Summary graphs for neuron density and cortical thickness. Data are means ± SEMs ('$n$' denotes the number of mice; neuron density: Ctrl and Nestin-*Slitrk2*, $n = 4$; cortical thickness: Ctrl and Nestin-*Slitrk2*, $n = 6$; two-tailed Mann–Whitney $U$ test). **j** Representative images of GAD67 (GABAergic interneuron marker) staining in the hippocampal CA1 of Nestin-*Slitrk2* mice. Scale bar: 0.1 mm. **k** Summary data for GAD67 staining from (**j**). Data are means ± SEMs ($n = 3$ mice each after averaging data from 3 sections/mouse; two-tailed Mann–Whitney $U$ test). **l, m** Representative immunoblots (**l**) of hippocampal lysates, and summary data showing synaptic protein levels (**m**) in crude synaptosomal fractions of P42 control and Nestin-*Slitrk2* brains, analyzed by semi-quantitative immunoblotting. Data are means ± SEMs ($n = 4$ mice/group; *$p < 0.05$, ***$p < 0.001$; two-tailed unpaired t-test). See Source data for raw data values and Supplementary Table 4 for statistical details.

(Fig. 7a, b). To corroborate these observations, we examined whether a SLITRK2 deletion affects the levels of TrkB-FL and TrkB-T in hippocampal CA1 lysates (Fig. 7c, d). We found that a deletion of *SLITRK2* induced a significant increase in TrkB-FL and decrease in TrkB-T (Fig. 7c, d), confirming the SLITRK2 gain-of-function effects. These results underscore the contribution of a normal

BDNF-TrkB activation range to the regulation of SLITRK2-mediated excitatory synaptic functions.

The accumulation of misfolded proteins is well known to trigger cellular stress responses, such as the endoplasmic reticulum (ER) stress response and unfolded protein response[30–32]. Thus, we hypothesized that expression of a subset of *SLITRK2* variants (i.e.,

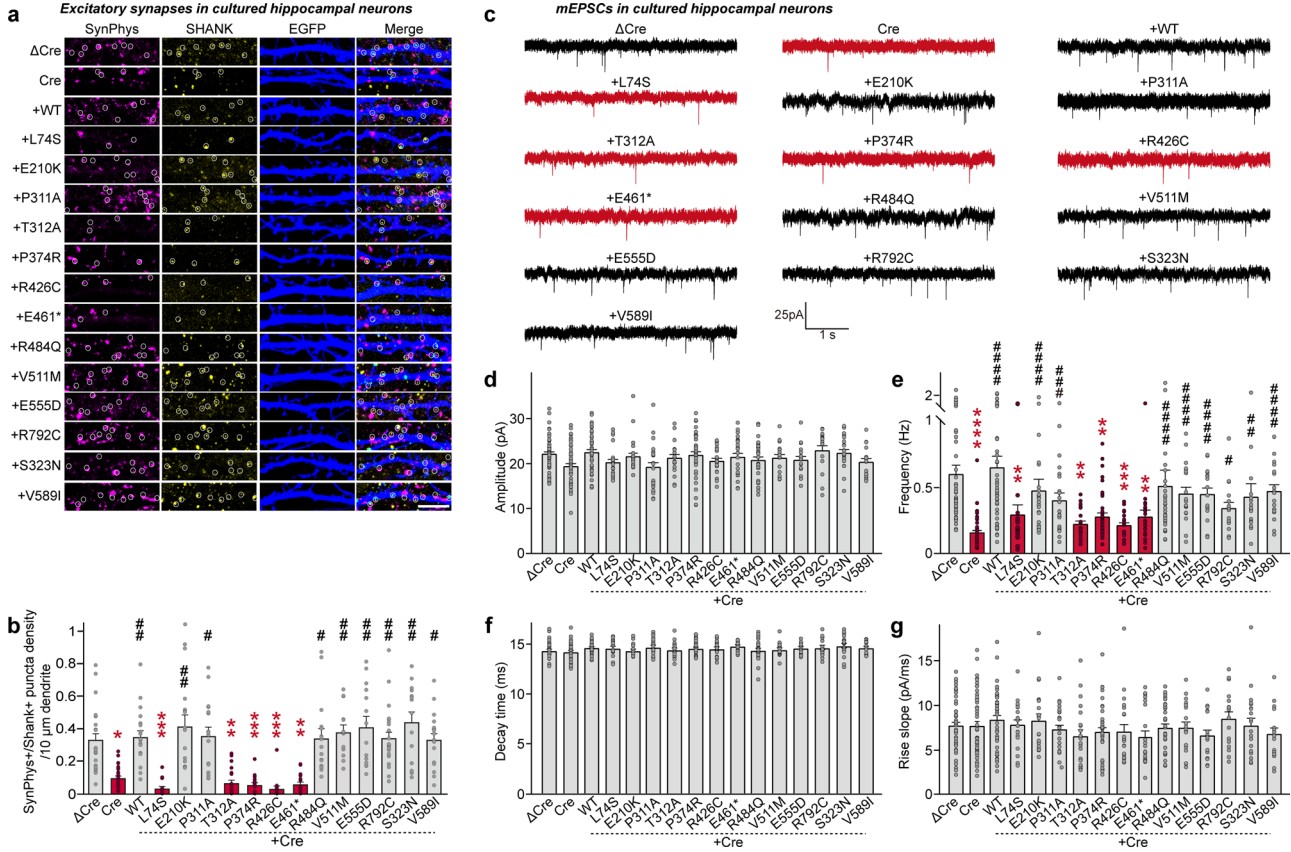

**Fig. 6 Impaired rescue of excitatory synapse development and transmission in *Slitrk2*-cKO hippocampal neurons by a subset of SLITRK2 variants.**
**a** Triple-immunofluorescence analysis of EGFP (blue), the presynaptic marker synaptophysin (SynPhys; magenta), and the excitatory postsynaptic marker SHANK (yellow) in mature cultured neurons (DIV14) derived from Slitrk2$^{f/f}$ mice infected with lentiviruses expressing ΔCre or Cre at DIV5. For rescue experiments, neurons infected with lentiviruses expressing Cre at DIV5 were transfected with vectors expressing WT or the indicated mutant forms of SLITRK2 together with EGFP vector at DIV8–9. Scale bar: 10 μm (applies to all images). **b** Quantification of images in (**a**), showing the density of SynPhys $^+$SHANK$^+$ puncta. All data are shown as means ± SEMs ('n' denotes the number of cells from five independent experiments; Control vs. experimental group: ΔCre, n = 22; Cre, n = 25; +WT, n = 18; +L74S, n = 12; +E210K, n = 17; +P311A, n = 16; +T312A, n = 18; +P374R, n = 16; +R426C, n = 13; +E461*, n = 15; +R484Q, n = 16; +V511M, n = 15; +E555D, n = 14; +R792C, n = 21; +S323N, n = 14; +V589I, n = 17; *p < 0.05, **p < 0.01, ***p < 0.001; Cre vs. experimental group: #p < 0.05, ##p < 0.01; ANOVA with a non-parametric Kruskal–Wallis test). **c–g** Representative mEPSC traces (**c**) and quantification of amplitudes (**d**), frequencies (**e**), decay time (**f**) and rise time (**g**) of mEPSCs recorded from hippocampal cultured neurons infected with lentiviruses expressing ΔCre or Cre at DIV5 and transfected with vectors expressing WT or the indicated mutant forms of SLITRK2 together with EGFP vector. All data are shown as means ± SEMs ('n' denotes the number of cells from six independent experiments; Control vs. experimental group: ΔCre, n = 48; Cre, n = 46; +WT, n = 41; +L74S, n = 20; +E210K, n = 22; +P311A, n = 24; +T312A, n = 19; +P374R, n = 36; +R426C, n = 20; +E461*, n = 22; +R484Q, n = 29; +V511M, n = 19; +E555D, n = 16; +R792C, n = 16; +S323N, n = 20; +V589I, n = 17; **p < 0.01, ***p < 0.001, ****p < 0.0001; experimental group; Cre vs. experimental group; #p < 0.05, ##p < 0.01, ###p < 0.001, ####p < 0.0001; ANOVA with a non-parametric Kruskal–Wallis test). See Source data for raw data values and Supplementary Table 4 for statistical details.

L74S, T312A, P374R, and R426C) exhibiting entrapment in intracellular compartments could induce an ER stress response in cultured hippocampal neurons. However, immunoblot analyses using lysates of neurons overexpressing SLITRK2 WT or the indicated SLITRK2 variants showed no difference in the levels of BiP, which are activated transcriptionally by ER stressors, between SLITRK2 WT and mutant variants (Fig. S17c, d), suggesting no effects of these variants on cellular stress responses.

**Slitrk2-cKO mice exhibit memory deficits and abnormal gait.**
We then tested whether *Slitrk2*-cKO mice exhibited the behavioral features that manifest in patients with NDD and *SLITRK2* variants (see Table 1). To this end, we performed a battery of behavioral tests, including those measuring learning and memory, locomotion, anxiety, and motor coordination (Fig. 8 and Fig. S18). Ten-week-old male Nestin-*Slitrk2* mice displayed comparable exploratory behavior (assessed by open-field tests) (Fig. S18a, b),

sociability, and social novelty recognition memory (assessed by three-chamber tests) as littermate control mice, showing preference for the first novel mouse over an inanimate object (Fig. S18c, d) and preference for a novel stranger mouse over familiar mouse (Fig. S18e, f). However, Nestin-*Slitrk2* mice exhibited reduced anxiety-like behavior, as evidenced by increased time spent in open arms (with a similar number of entries into each open arm) in the elevated plus-maze test, without a change in time spent in the light compartment in the light-dark test (Fig. S18g–i). Moreover, Nestin-*Slitrk2* mice showed impaired spatial learning and memory (assessed by novel object-recognition tests), although spontaneous alternation in Y-maze tests was similar to that of littermate control mice (Fig. S18j–m). We further examined spatial reference memory, as evaluated by the Barnes maze test[33]. Both Control and Nestin-*Slitrk2* mice learned to locate the target hole during the course of the training period, as indicated by gradual reductions in the number of search errors, escape latency and total

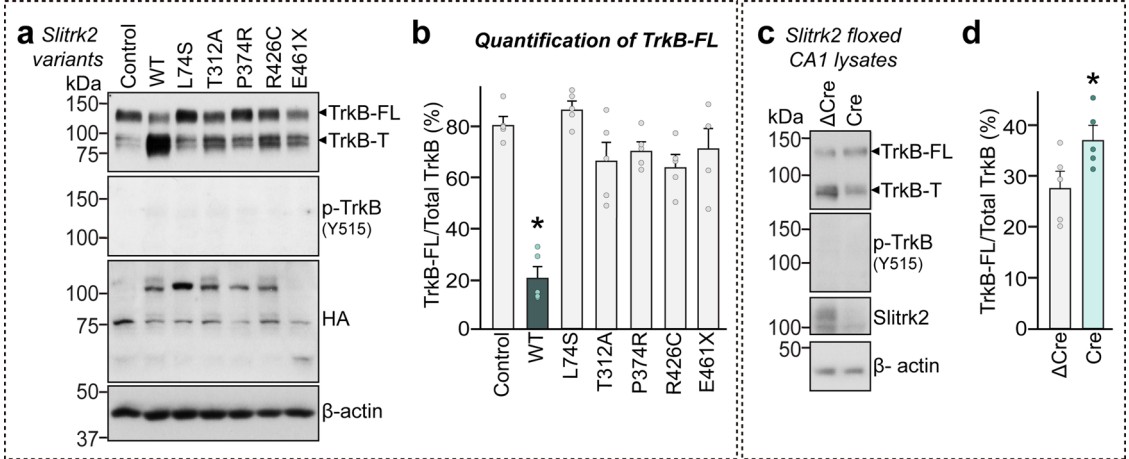

**Fig. 7 Alterations of full-length TrkB protein levels in cultured neurons by a subset of SLITRK2 variants.** Cultured cortical neurons were infected with lentiviruses expressing WT or the indicated mutant forms of SLITRK2. Levels of TrkB were examined by semi-quantitative immunoblotting using antibodies against TrkB or phospho-TrkB. FL full-length, T truncated. **b** Quantification of protein levels from (**a**). Summary data showing plots from five independent experiments. Data are mean ± SEMs (*$p < 0.05$; ANOVA with a non-parametric Kruskal–Wallis test). **c** Hippocampal CA1 lysates from Slitrk2-floxed mice infected with AAVs expressing ΔCre or Cre were examined by semi-quantitative immunoblotting using antibodies against TrkB or phospho-TrkB. FL full-length, T truncated. **d** Quantification of protein levels from (**c**). Data are means ± SEMs ($n = 5$ mouse samples; *$p < 0.05$; two-tailed unpaired $t$-test). See Source data for raw data values and Supplementary Table 4 for statistical details.

distance traveled, indicating normal acquisition of spatial reference memory in Nestin-*Slitrk2* mice (Fig. 8a–d). We then conducted the first probe test 3 days after the last day of training and the second probe test after 17 days. In the first probe test, the number of search errors was comparable between Control and Nestin-*Slitrk2* mice (Fig. 8e). In contrast, the accuracy of spatial memory of Nestin-*Slitrk2* mice in the second probe tests was worse than that of control mice, suggesting that retention of spatial reference memory is impaired in Nestin-*Slitrk2* mice (Fig. 8f).

Because some NDD patients reported in the current study suffer from spasticity and/or dystonia and unsteady gait (Table 1), we also examined whether Nestin-*Slitrk2* mice exhibit deficits in fine motor coordination. To this end, we analyzed the juvenile (P20) and adult (P65) Nestin-*Slitrk2* mice by measuring overlap distance, stance width, step angle, and gait symmetry. At both ages, the mean stride length, stance length, and sway length of forelimbs and hindlimbs of Nestin-*Slitrk2* mice were comparable to those of control littermate mice (Fig. 8g–i). Strikingly, Nestin-*Slitrk2* mice exhibited increased front/hind paw overlap, despite normal limb strength (assessed by grip strength test) (Figs. 8g–i, S18n). Taken together, these extensive behavioral data suggest that Nestin-*Slitrk2* mice exhibit impaired learning and memory and abnormal footprint patterns, partly recapitulating the clinical phenotypes of patients with *SLITRK2* variants (see Table 1).

**Correlation of impaired hippocampal CA1 excitatory synapse properties of *Slitrk2*-cKO mice with deficits in spatial reference memory.** Because Nestin-*Slitrk2* mice exhibited impairment in the Barnes maze test (Fig. 8a–f), a hippocampal CA1-dependent spatial task[34], we asked whether loss of Slitrk2 in the hippocampal CA1 region could recapitulate this behavioral deficit. To this end, we stereotactically injected adeno-associated viruses (AAVs) expressing Cre or ΔCre (Ctrl) into the CA1 of adult Slitrk2-floxed mice and performed semi-quantitative immunohistochemistry and electrophysiological recordings three weeks after injections (Fig. 9a, b). We found that the frequency, but not amplitude, of mEPSCs was markedly decreased in hippocampal CA1-specific *Slitrk2*-cKO mice (Fig. 9c–e). Moreover, these mice exhibited a significant reduction in the integrated intensity of VGLUT1 and PSD-95 puncta in the SO and SR layers (Fig. 9f, g), without

changes in VGAT puncta in any examined hippocampal CA1 layer (Fig. S19), in keeping with the anatomical phenotype of Nestin-*Slitrk2* mice (Fig. S16). Notably, co-expression of SLITRK2 WT, but not that of the SLITRK2 T312A variant, completely rescued the decreased integrated intensity of VGLUT1 and PSD-95 puncta in hippocampal CA1-specific *Slitrk2*-cKO mice (Fig. 9f, g). These results reinforce the phenotype of SLITRK2 T312A in hippocampal CA1 pyramidal neurons.

Finally, to determine whether CA1-specific elimination of Slitrk2 expression leads to long-term spatial memory deficits, we performed Barnes maze tests using CA1-specific *Slitrk2*-cKO mice. Similar to the behavioral phenotype of Nestin-*Slitrk2* mice, CA1 region-specific *Slitrk2*-cKO mice also exhibited impaired retention of spatial reference memory (Fig. 9h–l). Collectively, these results support the conclusion that SLITRK2 synaptic functions in the hippocampal CA1 are essential for execution of normal spatial tasks in mice.

## Discussion

Although numerous genes encoding synaptic cell adhesion proteins (CAMs) have been identified as causative factors for a variety of neurodevelopmental and/or neuropsychiatric disorders, it had remained to be determined whether a subset of neurological symptoms of patients with these disorders are caused by alterations in synaptic adhesion signaling pathways that are not clearly established. As a member of a vertebrate-specific synaptic cell adhesion protein family, SLITRK2 specifically functions in mediating proper excitatory synapse development[35]. In particular, SLITRK2 is required for organizing *trans*-synaptic signaling pathways involved in excitatory presynaptic assembly and postsynaptic maintenance by binding to PTPσ and PDZ domain-containing scaffolds[10,17,36]. In addition, our previous study demonstrated that a subset of disease-associated SLITRK2 missense variants induce impaired dendritic targeting of Slitrk2 in neurons, leading to dysfunctions of excitatory synapse maintenance[19]. The SLITRK2 V89M variant reduced excitatory synapse density, likely through its deleterious effects on surface expression and dendritic targeting[19]. Similarly, a series of other SLITRK variants associated with Tourette's syndrome, trichotillomania, schizophrenia, or OCD was shown to trigger loss of

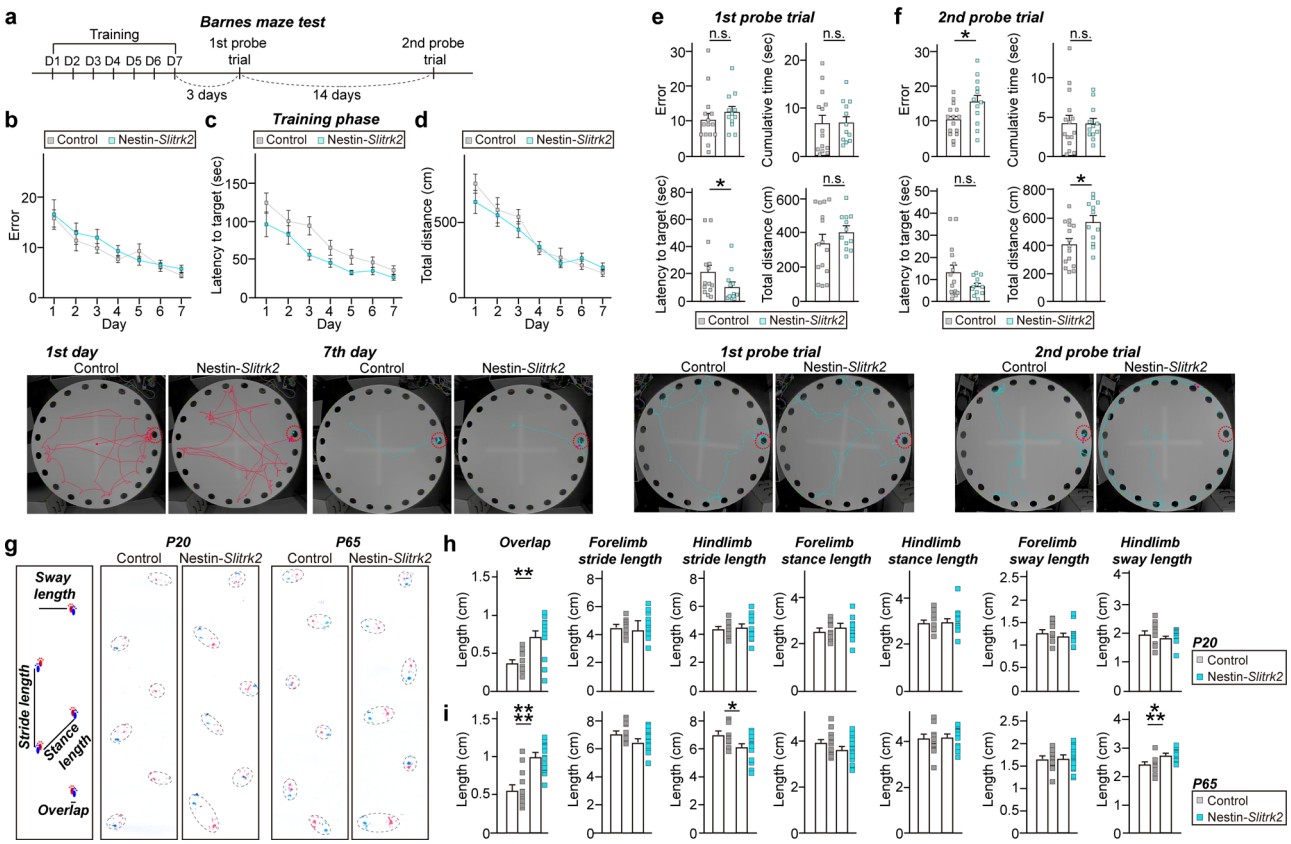

**Fig. 8 *Slitrk2*-cKO mice display mild learning and memory deficits and abnormal gait. a** Schematic depiction (top) of Barnes maze test. **b–d** Number of errors before first encountering the escape hole (**b**) escape latency (**c**) and total distance (**d**) for Control and Nestin-*Slitrk2* mice during the training session. Data are presented as means ± SEMs ('*n*' denotes the number of mice; Control, *n* = 17; Nestin-*Slitrk2*, *n* = 12; two-way ANOVA with Sidak's multiple comparisons test). Representative track images (bottom) from first and last days of the training. **e, f** Number of errors before first encountering the escape hole for control and Nestin-*Slitrk2* mice during 1st (**e**) and 2nd probe (**f**) trials. Data are presented as means ± SEMs ('*n*' denotes the number of mice; Control, *n* = 15; Nestin-*Slitrk2*, *n* = 12; *$p < 0.05$; two-tailed Mann–Whitney *U* test). Representative track images (bottom) from 1st and 2nd probe trials. **g** Representative image of the footprint patterns of juvenile (P20) and adult (P65) control and Nestin-*Slitrk2* mice. **h, i** Footprint patterns were analyzed based on overlap, stride length, stance length, and sway length. P20 (**h**) and P65 (**i**) Nestin-*Slitrk2* mice exhibited longer overlap length. Data are shown as means ± SEMs ('*n*' denotes the number of mice; P20: Control, *n* = 10; Nestin-*Slitrk2*, *n* = 12; P65: Control, *n* = 14; Nestin-*Slitrk2*, *n* = 15; *$p < 0.05$, **$p < 0.01$, ***$p < 0.001$, ****$p < 0.0001$; two-tailed Mann–Whitney *U* test). See Source data for raw data values and Supplementary Table 4 for statistical details.

the respective SLITRK protein functions in cultured hippocampal neurons[19,26]. These results collectively suggest that pathogenic *SLITRK* genetic variants likely induce loss-of-function of corresponding SLITRKs, accounting for the pathogenesis of patients with neurological disorders. However, previous studies, which primarily employed knockdown (KD) and/or overexpression approaches, preclude firm conclusions about SLITRK2 functions. Previous Slitrk2 loss-of-function studies performed in cultured neurons and mice produced contradictory results[10,17], with Slitrk2 KD in cultured neurons reducing excitatory synapse density, and Slitrk2 KD in hippocampal CA1 neurons failing to affect excitatory synapse density[10,17]. These results were previously attributed to differences in genetic manipulations, cellular preparation, or other unknown factor(s). Extensive analyses using *Slitrk2*-cKO mouse-derived cultured neurons recapitulated our previous anatomical results using Slitrk2 KD[10], in addition to exhibiting marked reduction in excitatory synaptic transmission. Moreover, *Slitrk2* deletion in adult mouse hippocampal CA1 neurons similarly reduces the frequency of mEPSCs, in contrast to the absence of an effect of global Slitrk2 KD on mEPSCs[17]. These results again underscore the importance of confirmatory analyses using a sophisticated system and approaches to elucidate functions of synaptic proteins.

In the current study, we identified *SLITRK2* variants in NDD characterized by developmental delay and speech delay, mild to severe ID, and various neurological and behavioral comorbidities. Seven of the eight individuals with potential disease-causing variants had significant anxiety, and four patients were diagnosed with autism spectrum disorder. Other behavioral diagnoses include ADHD with executive difficulties, hyperactivity, or agitation. Several individuals had unsteady gait associated with spasticity and/or dystonia and three patients had seizures, of variable type, with the severity of ID ranging from mild to severe. Such clinical heterogeneity among individuals is often reported in NDDs associated with mutations in the same gene and even precisely the same mutation[37,38], suggesting that other genetic and/or environmental modifying factors might also be involved. *SLITRK2* variants have previously been reported in two females with schizophrenia[18], but no patients affected by ID and/or NDD phenotypes and carrying *SLITRK2* pathogenic variants have been reported to date. Of the seven potential disease-causing variants identified in this study, five induced prominent alterations in the properties of SLITRK2. Among these, three variants (L74S, P374R, and R426C) perturbed surface transport of SLITRK2, accompanied by impaired binding to LAR-RPTPs and synaptogenic activity, similar to most of known *SLITRK1/4/5* substitutions that

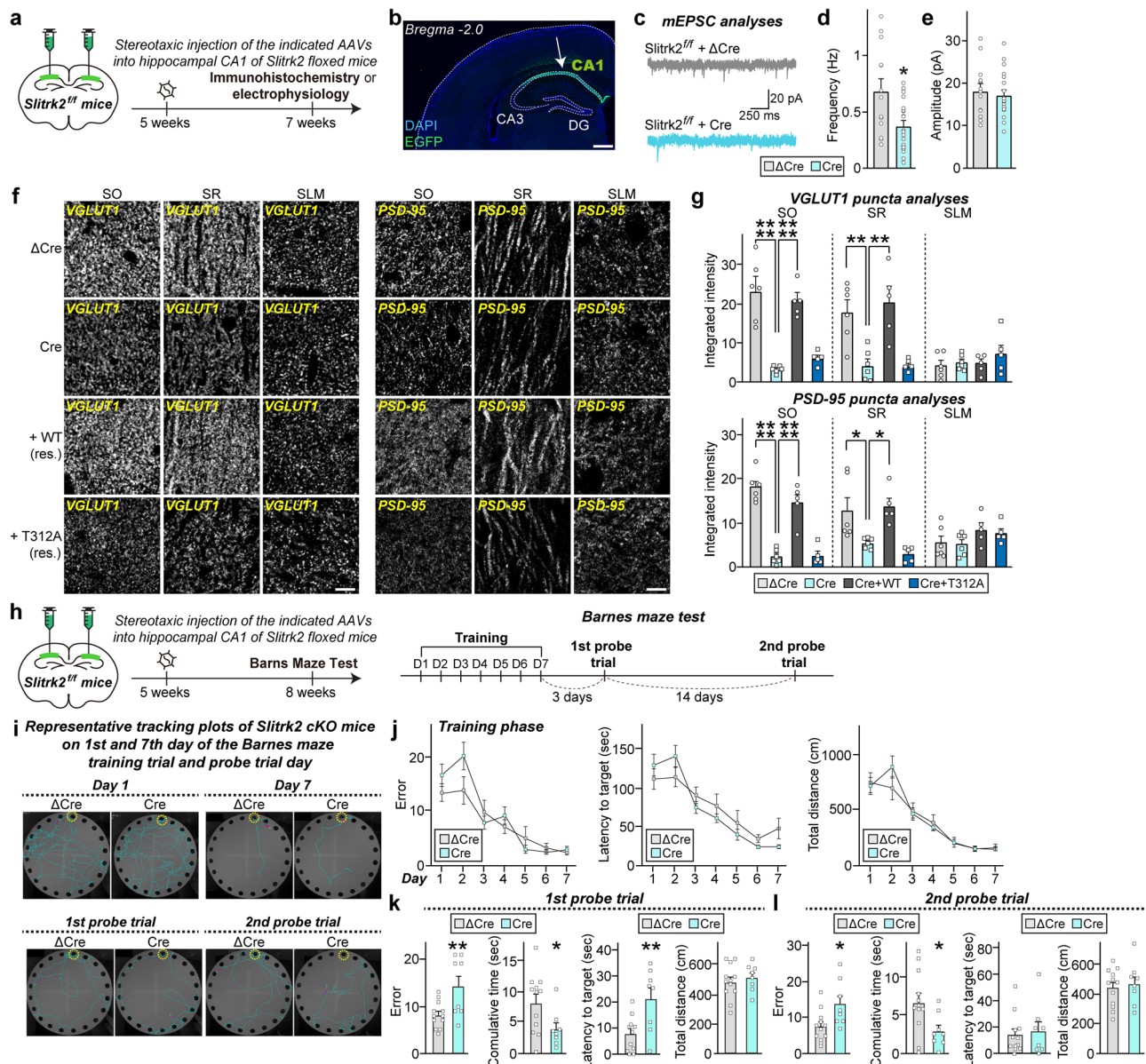

**Fig. 9 Proper Slitrk2 function in the hippocampal CA1 is required for mediating spatial reference memory in mice. a** Experimental configuration for electrophysiology and immunohistochemistry experiments (**c–g**). **b** Representative brain sections illustrating the precise targeting of AAVs for expression of Cre recombinase in the hippocampal CA1. Scale bar: 500 μm. **c–e** Representative traces (**c**) and summary graphs showing the frequency (**d**) and amplitude (**e**) of mEPSCs recorded from Slitrk2-floxed mice injected with AAVs expressing ΔCre or Cre. Data are shown as means ± SEMs (ΔCre, $n = 13$; Cre, $n = 20$; *$p < 0.05$; two-tailed Mann–Whitney $U$ test). **f, g** Representative images (**f**) and summary graphs quantifying the integrated intensity (**g**) of VGLUT1 and PSD-95 puncta. Data are shown as means ± SEMs (ΔCre, $n = 6$; Cre, $n = 6$; Cre+WT, $n = 5$; and Cre+T312A, $n = 5$ mice; *$p < 0.05$, **$p < 0.01$, ****$p < 0.0001$; ANOVA with Tukey's test). Scale bar, 10 μm (applies to all images). **h** Experimental configuration for Barnes Maze tests (**i–l**). **i** Representative traces of locomotive behavior for Barnes Maze tests. Target box is indicated with a yellow circle. **j** Number of errors before first encountering the escape hole, escape latency, and total distance for Slitrk2-floxed mice injected with ΔCre (Control) or Cre during the training session. Data are presented as means ± SEMs (ΔCre, $n = 12$; Cre, $n = 8$; two-way ANOVA with Sidak's multiple comparisons test). **k, l** Number of errors before first encountering the escape hole for Slitrk2-floxed mice injected with ΔCre (Control) or Cre during 1st (**k**) and 2nd (**l**) probe trials. Data are presented as means ± SEMs (ΔCre, $n = 12$; Cre, $n = 8$; *$p < 0.05$, **$p < 0.01$; two-tailed unpaired $t$-test). See Source data for raw data values and Supplementary Table 4 for statistical details.

represent loss-of-function mutations. One variant (E461*) in *SLITRK2* was a nonsense variant, and resulted therefore in a truncated SLITRK2 protein. The fifth variant, T312A, induced normal surface expression, and normal extracellular ligand-binding and synaptogenic activity, albeit with markedly reduced excitatory synapse number and excitatory synaptic transmission. Remarkably, these five SLITRK2 variants abrogated the ability of

SLITRK2 WT to negatively regulate TrkB-FL levels in neurons, revealing the significance of interactions between SLITRK2 and TrkB in terms of preventing SLITRK2 dysfunction-induced pathogenic effects. Given the well-established centrality of TrkB-mediated signaling in excitatory synapse development[39], it is plausible that selection of specific binding partners (i.e., LAR-RPTPs vs. TrkB) through fine regulation of SLITRK2-mediated

extracellular interactions constitutes an important signaling pathway for excitatory synapse organization. Our preliminary analyses suggested that Slitrk2, Slitrk3, and Slitrk5, but not other Slitrks, bind to TrkB (Fig. S20). Further experiments are warranted to dissect the molecular determinants underlying Slitrk2/5-TrkB interactions.

Our extensive behavioral analyses showed that Slitrk2 loss-of-functions in mice recapitulated a subset of behavioral symptoms found in patients with *SLITRK2* variants. Strikingly, the current study demonstrated that selective elimination of Slitrk2 expression in the hippocampal CA1 results in impaired retention of spatial reference memory. It is likely that specific alterations in synaptic inputs on CA1 neurons, possibly from either entorhinal cortex afferents or CA3 Schaffer-collateral afferents, are responsible for abnormalities in this behavioral task. Whether SLITRK2 is involved in the assembly of specific hippocampal circuits remains to be determined. In addition, Nestin-*Slitrk2* mice exhibited impaired long-term memory and altered footprint patterns. However, these mice displayed normal grip strength and somewhat enhanced motor coordination and learning. In general, the cerebellum is critical for coordinating movements, and its dysfunction often leads to uncoordinated walking or gait disturbance[40], a feature of patients harboring a subset of SLITRK2 variants, even though they were not clearly ataxic and the brain MRI did not show cerebellar anomalies (except for one patient with severe cerebral and cerebellar volume loss). Although the current study primarily correlated SLITRK2 synaptic functions in the hippocampal CA1 with spatial memory performance, future studies are warranted to identify the neuronal populations and/or neural circuits embedded in other brain areas that might underpin abnormal footprint patterns seen in Nestin-*Slitrk2* mice. Moreover, most patients harboring potential disease-causing variants in *SLITRK2* are anxious, whereas Nestin-*Slitrk2* mice appeared to exhibit reduced anxiety-like behavior in the elevated plus-maze tests. Future studies utilizing knock-in mice expressing disease-associated SLITRK2 variants should help resolve this issue using additional behavioral paradigms that examine anxiety-like behavior. Most patients exhibit seizures and speech delay. Thus, it will be necessary to clarify whether Nestin-*Slitrk2* mice exhibit seizure-like phenotypes or communication deficits by measuring electroencephalography and measuring ultrasonic vocalization, respectively. Because our various analyses did not reveal any prominent phenotypes in the case of E210K and V511M (predicted to be deleterious by in silico programs), their pathogenicity remains uncertain, although it is still possible that these variants are involved in SLITRK2-associated pathophysiology via as-yet unidentified mechanism(s) encompassing early brain developmental processes. Given that Slitrk2 is also expressed in specific astrocytic subtypes in adult mice[41], it is reasonable to speculate that these *SLITRK2* missense variants with no clear phenotypes in current functional assays might affect specific aspects of astrocytic functions that remain to be determined.

In conclusion, we demonstrated nonsense and missense variants in *SLITRK2* in a novel form of X-linked NDD characterized by different degrees of ID associated with motor, speech, and behavioral impairments, that are caused by perturbation of SLITRK2 function. Further studies with larger cohorts are needed to confirm the involvement of this gene in a NDD and better delineate the clinical manifestations of the latter, especially at the behavioral level.

## Methods

**Participants and genetic analysis.** Individuals were referred by clinical geneticists for genetic testing as part of routine clinical care. All patients enrolled and/or their legal representative signed informed consent for research use and authorization for publication. All the institutions received local institutional review board (IRB) approval to use these data in research. The main IRB approval was obtained from the Ethics Committee of the Strasbourg University Hospital (CCPPRB). Individuals P1 and P2 (male siblings) were followed up in the Department of Clinical Genetics of Montpellier University Hospital, and exome sequencing was performed for P1 in the Molecular Genetics Laboratory of Strasbourg University Hospital, France. The six remaining patients (P3–P10) were recruited through an international collaborative study and data-sharing through Genematcher[21]. All patients were followed by a referring clinical geneticist and underwent exome sequencing in the different molecular laboratories as detailed in Supplementary information. Sanger sequencing confirmed the *SLITRK2* variants in all affected probands and was used for segregation studies. Variants have been submitted to ClinVar database. The referring physicians completed clinical forms and sent morphological pictures and brain MRI reports, when available. Clinical information is detailed in Table 1. All patients' families provided written informed consent, and all procedures performed in the studies were done in accordance with the ethical standards of the local institutional research committee and the Declaration of Helsinki. Effects of *SLITRK2* missense variants were predicted using PolyPhen-2[42], SIFT[43], and CADD (combined annotation-dependent depletion) scores[44].

**Expression vectors.** pDisplay-SLITRK2 missense variants (L74S, V201I, E210K, P311A, T312A, S323N, P374R, R426C, R484Q, V511M, E555D, V589I, and R792C) were generated by polymerase chain reaction (PCR)-based mutagenesis with PrimeSTAR HS DNA polymerase (Takara) using a pDisplay SLITRK2 wild-type (WT) construct as a template. pDisplay-SLITRK2 E461* was generated by PCR amplification of the indicated region of human SLITRK2 (amino acid [aa] 22–460), followed by digestion with *Xma*I and *Sac*II and subcloning into a pDisplay vector (Invitrogen). pCMV5-SLITRK2 missense variants (S9I and G15E) were generated by PCR-based mutagenesis with PrimeSTAR HS DNA polymerase (Takara) using a pCMV5-WT SLITRK2-mVenus construct as a template. pCMV6-XL4-SLITRK2 variant-Fc constructs (L74S, P374R, and R426C) were generated by PCR-based mutagenesis with PrimeSTAR HS DNA polymerase (Takara) using the pCMV6-XL4-SLITRK2-Fc construct as a template. pCMV6-XL4-SLITRK2 E461*-Fc was generated by PCR amplification of the indicated region of human SLITRK2 (aa 22–460) using the pCMV6-XL4-SLITRK2-Fc construct as a template, followed by digestion with *Not*I and *Xba*I and subcloning into a modified pCMV6-XL4 vector in frame as a C-terminal Fc fusion protein. L-313 SLITRK2 missense variants were constructed by amplifying the indicated full-length SLITRK2 variants by PCR, digesting with *Nhe*I and *Bsr*GI, and subcloning the resulting products into the L-313 vector (lentiviral expression vector). pAAV-SLITRK2 T2A-GFP variants (WT and T312A) were generated by PCR amplification of the indicated region of human SLITRK2, followed by digestion with *Xba*I and *Bam*HI and subcloning into the pAAV-T2A-GFP vector. pDisplay-SLITRK3 and pDisplay-SLITRK5 were generated by PCR amplification of mouse Slitrk3 (aa 28–652) and human SLITRK5 (aa 41–664), respectively, followed by digestion with *Xma*I and *Sac*II and subcloning into the pDisplay vector. pCAGG-His-HA-Nlgn2 was generated by PCR amplification of rat Nlgn2 (aa 1-678) using pCMV5-Nlgn2 as a template, followed by digestion with *Eco*RI and *Not*I and subcloning into the pCAGG-His-HA vector. pCMV-SPORT6-TrkB was purchased from Korea Human Gene Bank. The following constructs were previously described: pDisplay-SLITRK2 WT[19], pCMV5-SLITRK2-mVenus[10], pDisplay-SLITRK1[10], pDisplay-SLITRK4[10], pCMV-IgC PTPδ[19], and pAAV-T2A-GFP[45]. FSW-ΔCre and FSW-Cre were gifts from Dr. Pascal S. Kaeser (Harvard University, Cambridge, MA, USA). pCMV6-XL4-SLITRK2-Fc was a gift from Dr. Davide Comoletti (Victoria University, New Zealand)[46].

**Antibodies.** A polyclonal rabbit antibody against SLITRK2 was generated using a fusion protein of glutathione S-transferase (GST) with human SLITRK2 (aa 705–784) produced in BL21 Escherichia coli and purified on a glutathione-Sepharose column (GE Healthcare). Following immunization of rabbits with GST-SLITRK2 [705-784], the SLITRK2-specific antibody, designated JK177 (RRID: AB_2892626), was affinity-purified using a GST-SLITRK2 [705-784]-immobilized Sulfolink column (Pierce). The following commercially available antibodies were used: mouse monoclonal anti-GM130 (clone 35/GM130; BD Transduction Laboratories; Cat# 610822; RRID: AB_398141), rabbit monoclonal anti-TfR (clone EPR20584; Abcam; Cat# ab214039; RRID: AB_2904534), guinea pig polyclonal anti-VGLUT1 (Millipore; Cat# AB5905; RRID: AB_2301751), mouse monoclonal anti-GAD67 (clone 1G10.2; Millipore; Cat# MAB5406; RRID: AB_2278725), mouse monoclonal anti-Synaptophysin (clone SVP-38; Sigma-Aldrich; Cat# S5768; RRID: AB_477523), mouse monoclonal anti-PSD-95 (clone K28/43; NeuroMab; Cat# 75-028; RRID: AB_2877189), rabbit polyclonal anti-GABA$_A$γ2 (Synaptic Systems; Cat# 224 003; RRID: AB_2263066), guinea pig polyclonal anti-VGAT (Synaptic Systems; Cat# 131 004; RRID: AB_887873), mouse monoclonal anti-HA (clone 16B12; BioLegend; Cat# 901501; RRID: AB_2565006), mouse monoclonal anti-MAP2 (clone AP-20; Sigma-Aldrich; Cat# M1406; AB_477171), rabbit polyclonal anti-MAP2 (Abcam; Cat# ab32454; AB_776174),rabbit monoclonal anti-TrkB (clone 80E3; Cell Signaling; Cat# 4603; RRID: AB_2155125), rabbit monoclonal anti-phospho-TrkB (Thermo Fisher; Cat# PA5-36695; RRID: AB_2553666), rabbit monoclonal anti-BiP (clone C50B12; Cell Signaling; Cat# 3177; RRID: AB_2119845), goat polyclonal anti-GFP (Rockland; Cat# 600-101-215; RRID: AB_218182), mouse monoclonal anti-GluN1 (clone 54.1; Millipore; Cat# MAB363; RRID: AB_94946), rabbit polyclonal anti-NR2A (Millipore; Cat# 07-632; RRID:

AB_310837), mouse monoclonal anti-β-actin (clone C4; Santa Cruz; Cat# sc-47778; RRID: AB_2714189), mouse monoclonal anti-NeuN (clone A60; Millipore; Cat# MAB377; RRID: AB_2298772), HRP-goat anti-human IgG crossed-adsorbed secondary antibody (Thermo Fisher; Cat# 62-8420; RRID: AB_2533962), Cy3-AffiniPure donkey Anti-rabbit IgG antibodies (Jackson ImmunoResearch; Cat# 711-165-152; RRID: AB_2307443), Cy3-AffiniPure donkey anti-mouse IgG antibodies (Jackson ImmunoResearch; Cat# 715-165-150; RRID: AB_2340813), Cy3-donkey anti-human IgG antibodies (Jackson ImmunoResearch; Cat# 709-165-149; RRID: AB_2340535), Cy3-donkey anti-guinea pig IgG antibodies (Jackson ImmunoResearch; Cat# 706-035-148; RRID: AB_ 2340447), FITC-AffiniPure donkey anti-mouse IgG antibodies (Jackson ImmunoResearch; Cat# 715-095-150; RRID: AB_2340792), FITC-AffiniPure donkey anti-goat IgG antibodies (Jackson ImmunoResearch; Cat# 705-095-147; RRID: AB_2340401), FITC-AffiniPure Donkey anti-rabbit IgG antibodies (Jackson ImmunoResearch; Cat# 711-095-152; RRID: AB_2315776), and goat anti-guinea pig IgG antibodies (Thermo Fisher; Cat# A-21450; RRID: AB_141882). The rabbit polyclonal anti-VGLUT1 (JK111; RRID: AB_2810945) and the rabbit polyclonal anti-PSD-95 (JK016; RRID: AB_2722693) were previously described[17]. Rabbit polyclonal anti-pan-Shank (1172; RRID: AB_2810261), rabbit polyclonal anti-GluA1 (1193; RRID: AB_2722772), and rabbit polyclonal anti-GluA2 (1195; RRID: AB_2722773) were gifts from Dr. Eunjoon Kim (KAIST, Korea) and previously described[51]. See Supplementary Table 3 for detailed information of antibody dilutions.

**Cell culture.** HEK293T cells (ATCC; CRL-3216) were cultured in Dulbecco's modified Eagle's medium (DMEM; Welgene) supplemented with 10% fetal bovine serum (FBS; Tissue Culture Biologicals) and 1% penicillin-streptomycin (Thermo Fisher) at 37 °C in a humidified 5% $CO_2$ atmosphere. All procedures were performed according to the guidelines and protocols for rodent experimentation approved by the Institutional Animal Care and Use Committee of Daegu Gyeongbuk Institute of Science and Technology (DGIST).

**Cell-surface binding assays.** Recombinant PTPδ Fc-fusion proteins were produced in HEK293T cells. HEK293T cells were transfected with a PTPδ-IgC construct or pCMV-IgC empty vector. After 72 h, the media from transfected cells were collected, and 50 mM HEPES (pH 7.4) and 0.5 mM EDTA were added. Soluble Fc-fusion proteins were purified using protein A-Sepharose beads (GE Healthcare). Pulled-down proteins were eluted with 0.1 M glycine (pH 2.2) and then neutralized with 1 M Tris-HCl (pH 8.0). HEK293T cells expressing HA-SLITRK2 and its variants were incubated with 10 μg/ml of the indicated PTPδ Fc-fusion proteins. Images were acquired using a confocal microscope (LSM800; Zeiss).

**Biotinylation assays.** HEK293T cells were transfected with HA-tagged SLITRK2 WT or the indicated missense variants. After 48 h, cells were washed twice with ice-cold phosphate-buffered saline (PBS), incubated with 1 mg/ml SulfoNHS-LC-biotin (Pierce) in ice-cold PBS for 30 min on ice, rinsed briefly once with 0.1 M glycine in PBS, and incubated with 0.1 M glycine/PBS for 10 min at room temperature to completely quench biotin reactions. Cells were lysed with lysis buffer (1% Triton X-100, 0.1% SDS, 50 mM Tris pH 7.4, 150 mM NaCl, and protease inhibitors) and incubated for 20 min on ice. After removal of cell debris by centrifugation, 250 μg of lysates was incubated with 30 μl streptavidin agarose beads (Pierce) for 4 h at 4 °C. After washing beads three times with lysis buffer, surface-labeled proteins were eluted with 2X sample buffer and analyzed by immuno-blotting using the indicated antibodies. Anti-transferrin receptor (TfR) and β-actin antibodies were used as surface and total protein loading controls, respectively.

**Staining for surface/intracellular protein levels.** HEK293T cells were transfected with expression vectors for HA-SLITRK2 WT or the indicated variants. After 48 h, cells were washed twice with PBS, fixed with 3.7% formaldehyde for 10 min at 4 °C, and blocked with 3% horse serum/0.1% bovine serum albumin (BSA; crystalline grade) in PBS for 15 min at room temperature. Surface-expressed SLITRK2 protein was then detected by staining with mouse anti-HA antibody at room temperature. After 90 min, cells were washed twice with PBS and incubated with FITC-conjugated anti-mouse antibodies for 1 h at room temperature. Cells were then permeabilized with PBS containing 0.2% Triton X-100 for 10 min at 4 °C and incubated with rabbit anti-HA antibody for 90 min at room temperature to label intracellularly expressed SLITRK2 proteins, followed by incubation with Cy3-conjugated anti-rabbit secondary antibodies.

**Production of recombinant viruses**
*Lentiviruses.* Lentiviruses were produced by transfecting HEK293T cells with three plasmids—lentivirus vectors, psPAX2, and pMD2.G—at a 2:2:1 ratio. After 72 h, lentiviruses were harvested by collecting the media from transfected HEK293T cells and centrifuging at $1000 \times g$ to remove cellular debris, as previously described[47].

*Adeno-associated viruses (AAVs).* For high-efficiency transfections, AAVs were packaged with pHelper and AAV1.0 (serotype 2/9) capsids, as previously described[48]. Briefly, HEK293T cells were co-transfected with pHelper and

pAAV10.2/9 vectors together with pAAV-hSynI-SLITRK2-T2A-EGFP, pAAV-hSynI-SLITRK2-T2A-EGFP (T312A), pAAV-hSynI-ΔCre-EGFP, or pAAV-hSynI-Cre-EGFP. Cells were harvested 72 h later, lysed, mixed with 40% polyethylene glycol and 2.5 M NaCl, and centrifuged at $2000 \times g$ for 30 min. The resulting pellets were resuspended in HEPES buffer (20 mM HEPES, 115 mM NaCl, 1.2 mM $CaCl_2$, 1.2 mM $MgCl_2$, 2.4 mM $KH_2PO_4$), mixed with an equal volume of chloroform, and centrifuged at $400 \times g$ for 5 min. The supernatants were concentrated three times with a Centriprep centrifugal filter (15 ml, 4310; Millipore) at $1220 \times g$ for 5 min each and then with an Amicon Ultra centrifugal filter (0.5 ml, 3 K MWCO; Millipore) at $16,000 \times g$ for 10 min. The infectious titer of viruses was assessed by qRT-PCR detection of EGFP sequences with subsequent reference to a standard curve generated using the pAAV-U6-EGFP plasmid.

**Animals.** All mice were on a C57BL/6J background, produced by back-crossing with C57BL/6J wild-type (WT) mice (purchased from Jackson Research Laboratory). The mice were kept and produced in the animal facility of Daegu Gyeongbuk Institute of Science and Technology under standard, temperature-controlled laboratory settings, including a 12-h light/dark cycle (lights on at 7 am) and free access to water and food. Pregnant Sprague-Dawley rats (Daehan Biolink) were used to prepare in vitro cultures of dissociated hippocampal neurons. All procedures were carried out in compliance with the guidelines and protocols for rodent experimentation in accordance with protocols (DGIST-IACUC-19052109-00) approved by the Institutional Animal Care and Use Committee of Daegu Gyeongbuk Institute of Science and Technology.

**Generation of *Slitrk2*-floxed mice.** The C57BL/6J mouse strain was used to produce our *Slitrk2*-cKO mice. All mice were housed in a specific pathogen-free facility.

*Targeting strategy.* The knockout strategy for conditional deletion of *Slitrk2* targeted exon 2–3 using the Cre-loxP system. Specifically, 5′ loxP was inserted into intron 1, and Frt-Neo-Frt-3′ loxP was inserted downstream of exon 3; mouse embryonic stem cell (mESC)/homologous recombination (HR) technology developed by Biocytogen was used to establish the *Slitrk2*-cKO mouse model.

*Donor vector construction.* The 5′ and 3′ homologous arms used in gene-targeting span ~3.5 kB and 3.0 kB, respectively. The targeting vector also expresses the diphtheria toxin A (DTA) gene as a negative screening tag, which can be used to deplete ESC clones with non-homologous recombination and improve ESC clone screening efficiency.

*Microinjection.* The targeting vector was linearized and electroporated into C57BL/6J mESCs. After G418 selection, a total of 1000 clones were selected in three separate rounds. Selected cell clones were verified by Southern blotting, karyotyping, and sequence validation of PCR products. Chimeric mice were generated by injecting positively selected C57BL/6J mESCs into a total of 80 (two rounds of 30 and 50) pre-implantation BALB/c mouse embryos (blastocysts).

*Southern blot analysis.* Chimeric offspring were crossed with WT C57BL/6 mice to produce the F1 generation. Germline transmission was then confirmed by Southern blotting and PCR analysis of tail DNA from F1 generation mice. The restriction enzymes *Eco*RI (5′ Probe-A), *Hind*III (Neo Probe (3′)), and *Sac*I (3′Probe-A) (all from New England Biolabs) were used for Southern blot analysis of genomic DNA. After digestion, genomic DNA extracted from mouse tails was separated on a 1% agarose gels and transferred to a positively charged nylon membrane (Hybond N+; Amersham International plc). The membrane was hybridized using DIG Easy Hyb Granules (Roche Applied Science Inc.) at 42 °C overnight in a mixture containing a PCR-generated probe labeled using the PCR DIG probe synthesis kit (Roche Applied Science Inc.). Hybridization signals were detected using the DIG Luminescent Detection Kit (Roche Applied Science Inc.).

**Quantitative RT-PCR in cultured neurons.** Cultured hippocampal and cortical neurons from *Slitrk2*-floxed mice were infected with ΔCre or Cre lentivirus at DIV4. Infected neurons were homogenized at DIV12 in TRIzol Reagent (Invitrogen), and total RNA was extracted according to the manufacturer's protocol. cDNA was prepared from total RNA using a cDNA synthesis kit (Takara Bio). qRT-PCR was performed on a CFX96 Real-Time PCR system (Bio-Rad) using TB Green reagents (Takara Bio) using a *Slitrk2*-specific probe (forward, 5′-GCA-GAGCTTGCAGTATCTCTATT-3′; reverse, 5′-GGACCTCAGCAGGTTGT-TATT-3′). A probe for *Gapdh* (forward, 5′-ACATGGTCTACATGTTCCAG-3′; reverse, 5′-TCGCTCCTGGAAGATGGTGAT-3′) was used as an endogenous control.

**Structural modeling.** The domain structures of human SLITRK2 (LRR1, aa C33-D270; LRR2, aa P341-P579) were modeled with AlphaFold2[22,23]. LRR1 and LRR2 domains of human SLITRK2 were presumed to be connected by a flexible linker based on previous negative-stain electron microscopy images of the full

ectodomain of human SLITRK1[14]. Protein stability of domain structures of human SLITRK2 and variants was calculated using FoldX 5 suite[49].

**Nissl staining**. Mice underwent intracardiac perfusion with PBS and then with 4% paraformaldehyde. Fixed brain tissue was isolated, post-fixed for 12 h at 4 °C, and dehydrated in 30% sucrose for 6–8 d. Then, brain tissue was embedded in OCT (Optimal Cutting Temperature) compound, frozen, cryo-sectioned at 20-μm thicknesses using a cryostat (Leica CM5120). Slices were mounted on glass slides, rinsed with PBS, and permeabilized with 0.1% Triton X-100 in PBS for 10 min. Permeabilized samples were washed twice with PBS (5 min each), then incubated for 20 min in 200 μl of NeuroTrace 500/525 Green Fluorescent Nissl Stain (Molecular Probes), diluted 1:100 in PBS before use. Following three more washes with PBS, the specimens were dried and mounted with Vectashield Mounting Medium containing 4′,6-diamidino-2-phenylindole (DAPI; Vector Laboratories). A slide scanner (Axio Scan.Z1; Carl Zeiss) was used to capture green fluorescence.

**Measurement of mouse ventricle volumes**. Acute brain slices (50 μm thickness) were obtained from 7-wk-old mice using a vibratome. Every sixth brain section (300 μm between sections) was collected starting from bregma +0.86 mm and continuing to the dorsal end at bregma −0.82 mm. Sections were immunostained with anti-NeuN (clone A60; Millipore), and whole-brain slice images were obtained using a slide scanner (Axio Scan.Z1; Carl Zeiss) with a ×20 objective lens. All image settings were kept constant. Lateral ventricle areas were manually sketched and calculated using MetaMorph software (Molecular Devices Corp.). Lateral ventricle volumes were calculated as the sum of sectional areas of lateral ventricles times the layer thickness.

**Heterologous synapse-formation assays**. Forty-eight hours after transfecting with the indicated expression vectors, HEK293T cells were trypsinized, seeded onto cultured hippocampal neurons at days in vitro (DIV)10, and cocultured for 48 h, as indicated. Cocultured neurons were co-immunostained with antibodies against the indicated proteins. Fluorescence images were acquired by confocal microscopy (LSM780, Carl Zeiss), and results were quantified by measuring the fluorescence intensities of synaptic marker puncta in randomly selected transfected HEK293 cells (region of interest), normalized with respect to the area of each cell. Results were quantified for both red and green channels using MetaMorph Software (Molecular Devices Corp.; RRID: SCR_002368).

**Primary neuronal culture, transfection, immunocytochemistry, and image acquisition and analysis**. Hippocampal and cortical mouse neuron cultures were prepared from embryonic day 17 (E17) *Slitrk2*-floxed mouse embryos, as previously described[50]. In brief, neurons were seeded on 18-mm poly-D-lysine (0.1 mg/ml)-coated coverslips and cultured in Neurobasal media (Gibco) containing penicillin-streptomycin and 0.5 mM GlutaMax (Thermo Fisher) supplemented with 2% B-27 (Thermo Fisher) and 0.5% FBS (Hyclone). Cultured hippocampal neurons were infected with lentiviruses expressing Cre or inactive Cre (ΔCre) recombinase at DIV5, and immunostained at DIV12 or DIV14. For rescue experiments, hippocampal neurons were transfected at DIV8–9 with SLITRK2 WT or the indicated point variants using a CalPhos Kit (Clontech) and immunostained at DIV13–14; controls were transfected with EGFP. For immunocytochemistry, cultured rat neurons were fixed with 4% paraformaldehyde/4% sucrose in PBS for 30 min at 4 °C and permeabilized with 0.2% Triton X-100 in PBS for 30 min at 4 °C. Neurons were blocked by incubating with 3% horse serum/0.1% BSA in PBS for 15 min at room temperature, then stained with the indicated primary and secondary antibodies for 70 min at room temperature. Z-stack images of randomly selected neurons were acquired using a confocal microscope (LSM780; Carl Zeiss) with a ×63 objective lens. Obtained Z-stack images were converted to maximal projections, and colocalized puncta densities were analyzed in a blinded manner using MetaMorph software (Molecular Devices Corp.).

**Sholl analysis**. Rat cultured hippocampal neurons were co-transfected with EGFP and SLITRK2 WT or the indicated SLITRK2 variants for 5 d at DIV9. Transfected neurons were stained with anti-EGFP and anti-HA antibodies, and costained neurons were used for quantitative analysis. Dendritic branching was analyzed by performing a Sholl analysis using the Scholl Analysis plugin installed in ImageJ software (Fiji, RRID: SCR_002285), as previously described[51]. The number of branches at increasing distances from the soma was quantified.

**Stereotactic surgery**. Mice were anesthetized with isoflurane (3–4%) and then placed in a stereotaxic device. Viral solutions (titers ≥1 × 10^11 viral genomes/ml) were injected bilaterally at a rate of 0.2 μl/min with a NanoFil syringe and a Nanoliter 2010 Injector (World Precision Instruments). The AP −2.5 mm, ML ± 1.7 mm, and DV −1.3 mm (from the dura) coordinates were used for hippocampal CA1 regions. Mice were used ≥3 weeks after stereotactic surgeries.

**Immunohistochemistry**. Mice were transcardially perfused first with PBS for 3 min and then with 4% paraformaldehyde in PBS for 5 min. After post-fixation

overnight, mouse brains were slowly sectioned at a thickness of 40 μm using a vibratome (VT1200S; Leica). Brain sections were permeabilized by incubation with 0.2% Triton X-100 in PBS containing 10% horse serum and 0.2% BSA for 1 h at room temperature. For immunostaining, sections were incubated overnight at 4 °C with primary anti-VGLUT1 (1:500), anti-PSD-95 (1:500), or anti-VGAT (1:500) antibodies diluted in the same blocking solution. After washing three times, sections were incubated with Cy3-conjugated secondary antibodies (Jackson ImmunoResearch) for 1 h at room temperature. Sections were then washed extensively and mounted on glass slides with Vectashield Mounting Medium (Vector Laboratories). Images were acquired by confocal microscopy (LSM700; Zeiss). Synaptic puncta were quantified using MetaMorph software (Molecular Devices), and their density and average area were measured.

**Mouse behavioral tests**. Male mice at the age of 6–10 weeks were used for all behavioral tests. Tests were conducted in a sound-proof room under dim lighting (<5 lux).

*Open-field test*. Each mouse was placed into an open-field apparatus (40 × 40 × 40 cm) and allowed to freely explore the field for 30 min. The total distance traveled and time spent in the center area were recorded by a top-view infrared camera and analyzed using EthoVision XT 10.5 (Noldus).

*Three-chamber test*. The testing apparatus consisted of a white acrylic box divided into three chambers (20 × 40 × 22 cm each) with small openings on the dividing walls. Wire cups were employed to enclose social conspecifics in the corners of both side-chambers. Mouse was placed in the central chamber for a 10-min habituation. Following habituation, an age-matched social conspecific was placed into the wire cup on the left side-chamber, and the sociability of the subjects was assessed by measuring subjects' exploration times for the enclosed conspecific and the empty cup during the second 10-min session. An exploration was counted when the subjects directed their nose into the vicinity of the wire cups. In the last 10-min session, a new social conspecific was placed into the empty wire cup on the right side-chamber, and social recognition was assessed by measuring subjects' exploration times for the familiar conspecific and the novel conspecific in a wire cup

*Elevated plus-maze test*. The white acrylic maze with two open arms (30 × 5 × 0.5 cm) and two closed arms (30 × 5 × 30 cm) was elevated 75 cm over the floor. Mice were individually placed at the center of the elevated plus-maze and allowed to freely move for 5 min. All behaviors were recorded by a top-view infrared camera, and the time spent in each arm and the number of arm entries were analyzed using EthoVision XT 10.5 (Noldus).

*Light and dark box test*. The apparatus consists of a roofless box divided into a closed dark chamber and a brightly illuminated chamber. A small entrance allows free travel between the two chambers. The light chamber was illuminated at 350 lux. The movement of mice was recorded by a top-view infrared camera and EthoVision XT 10.5 (Noldus) was used for analysis of the time spent in each chamber and the number of transitions

*Y-maze test*. Spatial working memory was evaluated on the basis of spontaneous alternation behavior in a Y-maze consisting of three 40-cm-long arms at a 120° angle from each other. Each mouse was placed at the center of the maze and allowed to explore freely for 8 min. An entry was counted when the center point of a mouse was within the arm. The movement of mice was recorded by a top-view infrared camera and analyzed using EthoVision XT 10.5 (Noldus).

*Novel object-recognition test*. Object recognition test was performed in the open-field box. On the first day, mice were habituated to the maze for more than 10 min. On the subsequent day, mice were allowed to explore two identical objects twice for 10 min. Exploration time for each object was measured. Twelve hours later, one of two objects was replaced with a new distinct object and mice were returned to the field and allowed to explore for 10 min to test novel-object recognition. Exploration time for each object was measured.

*Barnes maze test*. The Barnes maze test was conducted on a white circular platform, 95 cm in diameter, with 20 holes equally spaced around the perimeter. The hole above the escape box represented the target. The location of the target was consistent for a given mouse but randomized across mice. The maze was rotated daily, with the spatial location of the target unchanged with respect to the distal visual room cues. To prevent a bias based on olfactory or proximal cues within the maze. Three trials per day were conducted for 7 successive days. On day 10, the first probe trial was conducted without the escape box to confirm that the spatial task was acquired based on navigation by distal environment room cues. On day 17, the second probe trial was conducted. During acquisition and probe trials, the following parameters were measured by a top-view infrared camera and analyzed using EthoVision XT 10.5 (Noldus): velocity, error score, total distance, first visit time to escape box, and total latency.

*Grip strength test*. A grip strength meter (BIOSEB) was used to measure forelimb grip strength. A mouse was allowed to grasp the steel bar mounted on the force gauge. The gauge was reset after stabilization, and the mouse's tail was slowly pulled back by an experimenter. Maximal grip force was automatically recorded as newtons by the gauge. We performed 3 consecutive measurements per day for 3 consecutive days.

*Gait test*. The forepaws and hind paws of a mouse were coated with different colored nontoxic acrylic paints. The mice were trained to run down the runway (30-cm long and 5-cm wide) in a straight line before the test. The footprints were traced on the white paper covering a runway. The footprints were analyzed for overlap, each side of stride length, each side of stance length, and each side of sway length. All parameters were analyzed by MetaMorph software (Molecular Devices Corp.).

### Electrophysiology

*Cultured neuron electrophysiology*. Hippocampal cultured neurons obtained from *Slitrk2*-cKO mice were infected on DIV4 with lentiviruses encoding Cre-EGFP or ΔCre-EGFP, followed by analysis at DIV13–16. Similarly, hippocampal cultured neurons obtained from Nestin-*Slitrk2* mice were analyzed at DIV13–16. Pipettes were pulled from borosilicate glass (o.d. 1.5 mm, i.d. 0.86 mm; Sutter Instruments) using a Model P-97 pipette puller (Sutter Instruments). The resistance of patch pipettes filled with internal solution varied between 3 and 6 MΩ. The internal solution contained 145 mM CsCl, 5 mM NaCl, 10 mM HEPES, 10 mM EGTA, 0.3 mM Na-GTP and 4 mM Mg-ATP, with pH adjusted to 7.2–7.4 with CsOH and an osmolarity of 290–295 mOsmol/l. The external solution consisted of 130 mM NaCl, 4 mM KCl, 2 mM CaCl$_2$, 1 mM MgCl$_2$, 10 mM HEPES, and 10 mM D-glucose, with pH adjusted to 7.2–7.4 with NaOH and an osmolarity of 300–305 mOsmol/l. The whole-cell configuration was obtained at room temperature using μM-TSC manipulators (SENSAPEX). Electrophysiological data were acquired with a Multiclamp 700B amplifier (Axon instrument) and pCLAMP software, and were digitized using an Axon DigiData 1550B data acquisition board (Axon Instruments). mEPSCs were recorded at a holding potential of −70 mV. Synaptic currents were analyzed offline using Clampfit 10.8 software (Molecular Devices). For mEPSC recordings, the external solution contained 1 μM TTX and 50 μM picrotoxin to block GABA$_A$ receptor and Na$^+$ currents, respectively. For mIPSC recordings, the external solution contained 1 μM TTX, 10 μM CNQX, and 50 μM D-AP5 to block Na$^+$ currents, AMPA receptors, and NMDA receptors, respectively.

*Hippocampal CA1 pyramidal neuron electrophysiology*. Whole-cell voltage-clamp recordings were obtained from acute brain slices. Brain slices were transferred to a recording chamber and perfused with a bath solution of aerated (O$_2$ 95%/CO$_2$ 5% mixed gas) artificial cerebrospinal fluid (aCSF) consisting of 124 mM NaCl, 3.3 mM KCl, 1.3 mM NaH$_2$PO$_4$, 26 mM NaHCO$_3$, 11 mM D-glucose, 2.5 mM CaCl$_2$, and 1.5 mM MgCl$_2$ at 28–30 °C. For measuring postsynaptic currents, patch pipettes (open pipette resistance, 3–5 MΩ) were filled with an internal solution consisting of 145 mM CsCl, 5 mM NaCl, 10 mM HEPES, 10 mM EGTA, 4 mM Mg–ATP, and 0.3 mM Na–GTP. Whole-cell recordings of mEPSCs were performed on CA1 pyramidal neurons, voltage clamped at −70 mV, and currents were pharmacologically isolated by bath applications of 50 μM picrotoxin (Tocris) and 1 μM tetrodotoxin (Tocris). Electrophysiological data were acquired using pCLAMP software and a MultiClamp 700B (Axon Instruments) and were digitized using an Axon DigiData 1550B data acquisition board (Axon Instruments). Data were sampled at 10 kHz and filtered at 4 kHz. Data were discarded if the series resistance was >30 MU, or the series resistance differed by more than 20%.

### Data analyses

Data analysis and statistical tests were performed using GraphPad Prism8.0 software (RRID: SCR_002798). Heterologous synapse-formation assays and surface-binding assays were quantified by randomly selecting transfected HEK293T cells as the region of interest. The fluorescence intensities of synaptic marker puncta or Fc-fusion proteins were normalized to transfected protein signal intensities using MetaMorph software (Molecular Devices Corp.). All data are expressed as means ± standard error of the mean (SEM) unless stated otherwise, and significance is indicated with an asterisk. All experiments were performed using at least three independent mice, cultures, and/or cohorts of grouped mice, and the normality of data distributions was evaluated using the Shapiro–Wilk test. Data were compared using Student's *t*-test or one-way analysis of variance (ANOVA) using a non-parametric Kruskal–Wallis test, followed by Dunn's multiple comparison test for post hoc group comparisons, *t*-test, Mann–Whitney *U* test, or Fisher's least significance difference; 'n' values used are indicated in the figure legends, and numbers shown indicate replicates. Tests used to determine statistical significance are stated in the text and legends of figures depicting the results of the respective experiments. A *p*-value <0.05 was considered statistically significant, and individual *p*-values are indicated in the respective figure legend. All experiments were performed and analyzed in a blind manner by the experimenter.

**Reporting summary**. Further information on research design is available in the Nature Research Reporting Summary linked to this article.

## Data availability

Data sets presented in this study are included in full wherever possible, including the display of individual data points. All relevant data supporting the findings of this study are available from the corresponding authors upon reasonable request. Biological materials, including mutant mice and custom antibodies generated for this study, will be shared upon request within the limits of respective material transfer agreements for as long as they are available in the laboratory. Source data are provided with this paper.

## Code availability

All relevant code supporting the findings of this study is available from the corresponding authors upon request.

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

## Acknowledgements

We thank Jinha Kim (DGIST, Korea) for technical assistance, Vaidehi Jobanputra (Columbia University, USA) and Lenet Watt Skovstrøm (Amplexa Genetics, Denmark) for genomic analysis. This study was supported by grants from the National Research Foundation of Korea (NRF) funded by the Ministry of Science and Future Planning (2019R1A2C1086048 and 2020R1A4A1019009 to J.W.U.; 2021R1A2C1091863 to J.Ko.), the DGIST R&D Program of the Ministry of Science and ICT (22-CoE-BT-01 to J.Ko. and J.W.U.), Institute for Basic Science (IBS-R030-C1 to H.M.K.), NIH (NS106298 to M.C.K.), and SFARI and JPB Foundation (to W.K.C.).

## Author contributions

Conceptualization: J.Ko., A.P., and J.W.U. Methodology: K.A.H., D.K., G.J., D.L., H.Y.K., J.Kim., and A.r.H. Formal analysis: A.P., S.B, M.C.K., J.W., W.K.C., G.V., I. C., M.P., J.G., K.H, M.W., A.B., A.F.J., A.S., S.M., A.P.M.d.B., A.V.S, M.A., J.S., S.K., B.I., B.C., M.N., C.F., J.M., E.T., D.K.G., and M.W. Writing: original draft, S.E.C., J.Ko., A.P., and J.W.U.; review and editing, S.E.C., K.A.H., H.M.K., J.Ko., A.P., and J.W.U. Supervision: J.Ko., A.P., and J.W.U. Funding acquisition: H.M.K., M.C.K., W.K.C., J.Ko., and J.W.U.

## Competing interests

The authors declare no competing interests.

## Additional information

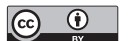

[1]Service de Génétique Médicale, Institut de Génétique Médicale d'Alsace (IGMA), Hôpitaux Universitaires de Strasbourg, Strasbourg, France. [2]Institut de Génétique et de Biologie Moléculaire et Cellulaire (IGBMC), INSERM U1258, CNRS-UMR7104, Université de Strasbourg, Illkirch-Graffenstaden, France. [3]Laboratoire de Génétique Médicale, UMRS_1112, Institut de Génétique Médicale d'Alsace (IGMA), Université de Strasbourg et INSERM, Strasbourg, France. [4]Department of Brain Sciences, Daegu Gyeongbuk Institute of Science and Technology (DGIST), Daegu 42988, Korea. [5]Pediatric Movement Disorders Program, Division of Pediatric Neurology, Barrow Neurological Institute, Phoenix

Children's Hospital, Phoenix, AZ, USA. [6]Departments of Child Health, Neurology, Cellular & Molecular Medicine and Program in Genetics, University of Arizona College of Medicine, Phoenix, AZ, USA. [7]Departments of Pediatrics, Columbia University Medical Center, New York, NY, USA. [8]Department of Medicine, Columbia University, New York, NY 10032, USA. [9]Department of Molecular Medicine and Medical Biotechnologies, Federico II University Hospital, Via Pansini 5, 80131 Naples, Italy. [10]Translational Medicine, UCB Pharma, Slough, UK. [11]Adelaide Medical School, Faculty of Health and Medical Sciences, The University of Adelaide, Adelaide, South Australia, Australia. [12]Robinson Research Institute, The University of Adelaide, Adelaide, South Australia, Australia. [13]Women and Kids, South Australian Health and Medical Research Institute, Adelaide, South Australia, Australia. [14]Center for Biomolecular and Cellular Structure, Institute for Basic Science, Daejeon 34126, Korea. [15]Graduate School of Medical Science and Engineering, Korea Advanced Institute of Science and Technology (KAIST), Daejeon 34141, Korea. [16]Department of Clinical Genetics, Erasmus University Medical Center, Rotterdam, The Netherlands. [17]Department of Epilepsy Genetics and Personalized Medicine, Danish Epilepsy Center, Dianalund, Denmark. [18]Institute for Regional Health Services, University of Southern Denmark, Odense, Denmark. [19]Department of Pediatrics Neurology, Quirónsalud Hospital & Universidad Europea, Madrid, Spain. [20]Department of Pediatrics, Center for Fragile Child, ASST Lariana Sant'Anna Hospital, San Fermo della Battaglia, Como, Italy. [21]Fondazione MBBM, Monza, Italy. [22]Department of Human Genetics, Donders Institute for Brain, Cognition and Behavior, Radboud University Medical Center, Nijmegen, The Netherlands. [23]Department of Clinical Genetics, Maastricht University Medical Centre, Maastricht, The Netherlands. [24]Translational Medicine, UCB Pharma, Braine-l'Alleud, Belgium. [25]Paediatric Neurosciences Research Group, Royal Hospital for Children, Queen Elizabeth University Hospitals, Glasgow, UK. [26]Service de Génétique Médicale, CHU Nantes, Nantes, France. [27]Nantes Université, CNRS, INSERM, l'institut du thorax, F-44000 Nantes, France. [28]Laboratoire de Diagnostic Génétique, Institut de Génétique Médicale d'Alsace (IGMA), Hôpitaux Universitaires de Strasbourg, Strasbourg, France. [29]GeneDx, Gaithersburg, MD 20877, USA. [30]Division of Genetics and Genomic Medicine, Department of Pediatrics, Washington University School of Medicine, St Louis, MO, USA. [31]Service de Génétique Médicale, Reference Centre AD SOOR, AnDDI-RARE, Inserm U1298, INM, Arnaud de Villeneuve Hospital and University of Montpellier, Montpellier, France. [32]Institut Universitaire de France, Paris, France. [33]These authors contributed equally: Salima El Chehadeh, Kyung Ah Han, Dongwook Kim, Gyubin Jang. ✉email: jaewonko@dgist.ac.kr; piton@igbmc.fr; jiwonum@dgist.ac.kr

