## [Peer Review File · Nature Communications]

REVIEWER COMMENTS

Reviewer #1 (Remarks to the Author):

This is the Functional characterization of several patient mutations in SLITRK2 identified by human genetics studies

Cellular and molecular characterization combined with behavioral and electrophysiological studies show differences among how different patient mutations could affect differently brain function make this study highly relevant and broad significance to the fields of neurodevelopmental disorders, neurotrophin, and human genetics field. While the role of SLITRKs in synaptogenesis has been reported before this work is really the first that does such an extensive study in terms of carefully characterizing the effect of different disease variants therefore it has broad appeal.

The work supports the authors claims and conclusions at large, but there are some specific points that I point out below that need to be revised and further supported as I think there are some experiments that require revisions and need to be expanded to solidify their conclusions

Sufficient detail is provided for the different experiments presented

Specific comments per figure:

Figure 1 - can the authors show sequencing traces showing the mutations maybe put that in the supplementary material?

Figure 2 - minor point make the quadrants bigger are all at the same scale?

Figure 3C this is a very nice experiment showing changes in surface localization have they tried staining for other membrane markers for the mutants that are changing to further confirm /validate the changes in surface localization?

Figure 4

In heK cells the ones that are changing are L74S, P374R, p426C, E361*Those are the ones that are also changing in the neurons (WT)

But have they looked at the variants that are changing and use them to try to rescue the KO neurons that would be even more telling or further proof of what they are seeing by overexpression

Is it possible to make less dense cultures or take images of neurons that are in regions where there are less neurons wondering if the lack of HA signal in the dendrites is because the arbors are reduced on those neurons

Also have they try staining for golgi, er markers on the neurons in which they express the different pathogenic variants to show the mechanism of retention in the cell bodies?

Figure 5

Its very clear that slktr2 is pretty much gone and that there is a defect that is very specific to glua2 in terms of protein levels and while there does not seem to be a change in numbers of neurons at the hippocampus from the sections the authors show there seems to be thinning of the cortex. Have they measured this? Or weigh the brains to see if compared to body weights there is a difference?

Since Nestin CRE will target not only the progenitors for excitatory neurons but also to inhibitory interneurons have they look to see if there are any changes in inhibitory interneurons (numbers of cells or marker protein levels?)

Figure 6

Why was this experiment done using the CRE virus on the primary cultures of the SLITRK2 flox mice? Why did the authors did not just harvest the neurons from the SLITRK2 flow/nestin CRE animals and use that directly? Wouldn't there be potential differences due to a more accute knockout on the cultures with the experiment they use?

Also in panel B they quantify the density the effect is clear from looking at the images the puncta in some of the mutants seems to be larger have the measure the size of the puncta? have they measure the size of the puncta?

For panel E could the authors have a supplementary table showing the statistics for all the comparisons, some of the mutants that are compare to the CRE seem to have a smaller frequency compared to the control (delta CRE) so would be good to be able to see if they are in any way different from the delta CRE

Figure 7 their evidence for the TrkB connection needs to be expanded to and made clearer in order to solidify these data

- First they need to also look at the other 2 mutants in which they see a phenotype to see if the effect is equal across mutations
- Second they need to look at phosphorylation of trkB as a ratio of total trkB from the blot it seems they are only looking at total trkB
- Third, Since there are other isoforms of TrkB like TrkB -SHC they have to clarify why they are not looking into these other isoforms

Figure 8 - interesting behavioral data could the authors specify if the studies were done only in males since the condition is X-linked?

Figure S3, would be good to have an internal control for lysates and if possible for secreted proteins while the effect is clear it would be good to have the values normalized to internal protein control and then that number normalized to the WT levels

Also did they discuss why is E461* so different compared to the other 2 mutants?

Figure S5, this is a very interesting finding using the heterologous system can the authors analyze inhibitory synapses in the KO animals? Maybe puncta of GAD67?

Is the lack of dendritic localization of SLITRK pathogenic variants due to

Overall this is a very exciting study that includes human genetics, mouse behavior and cellular characterization on the function of SLITRK2 disease variants.

Reviewer #2 (Remarks to the Author):

In this manuscript, Chehadeh et al. report a number of new missense variants that further implicate the transmembrane synaptic adhesion protein Slitrk2 in the pathogenesis of X-linked intellectual disability. The authors examine the biochemical interactions, cell surface transport, and synapse forming activities of these missense mutants using mainly in vitro approaches, and they also perform behavioral analysis of newly generated conditional Slitrk2 knock out mice. The paper is ambitious, technically rigorous, and contributes significantly to our understanding of the genetic basis of X-linked intellectual disability (XLID) and related neurodevelopmental disorders. It also reports the production of two valuable research reagents in the synaptic adhesion research community for which no commercially reliable products exist

– a Slitrk2 antibody, as well as a Slitrk2 conditional knock out mouse. The manuscript is remarkably well written – both comprehensive and easy to follow. I have only three moderate concerns and a few minor suggestions to communicate.

Major points:

1. In Figure 3, the authors claim that three missense variants (L74S, P374S, and R426C) do not exhibit the cell surface expression that is characteristic of WT Slitrk2 and other missense variants. However, upon examination of the images in Figure 3C, it is not entirely convincing that the staining patterns observed for these mutants are intracellular. A similar concern is applicable to Supplementary Figure 5, where, again, the staining patterns of these mutants appears more surface than intracellular. Because this claim of a surface transport defect is an important part of the paper's premise, additional surface vs. intracellular data are warranted to strengthen this claim.

2. In Figure 6, as well as Supplementary Figure 5, it is curious that the authors only report data on excitatory synapse formation, while omitting any analysis of inhibitory synapse formation. There is some suggestion in previous literature (see Takahashi et al. 2012, Figure 1) that Slitrk2 can induce the formation of both excitatory and inhibitory synapses in a co-culture assay. Given the uncertainty regarding Slitrk2's ability to induce inhibitory synapses, the inclusion of inhibitory synapse markers in these experiments would be informative.

3. In the discussion, the authors state that they have found an association of Slitrks 2, 3, and 5 with TrkB, but no more information is given, and no data is shown. Given that the TrkB interaction forms the basis of Figure 7, the inclusion of these data would be a valuable addition to Figure 7, or at least as a figure in the supplementary materials.

Minor points:

In Figure 3B, it appears at least by banding pattern that some of the mutants have altered glycosylation or other posttranslational differences. Has this been examined experimentally?

In Figure 4, it is interesting that dendritic trafficking of each the four featured mutants is compromised, yet the relative amount of protein in the somatic compartment is not elevated. If expression levels of these mutants of interest are unchanged (Fig 3A) in comparison to other forms of Slitrk2, then it would seem that somatic measurements of these mutants should be elevated.

With regard to the mention of PDZ-containing scaffolds in Line 652 of the discussion, Loomis et al. 2020 should also be included as a reference.

All figures containing immunocytochemistry should be reformatted so as to avoid the red/green color scheme.

Reviewer #3 (Remarks to the Author):

In this manuscript, the authors identify novel mutations in the SLITRK2 gene associated with X-linked neurodevelopmental disorders. They demonstrate that several of these mutations lead to a loss in Slitrk2 function at synapses due to lack of receptor expression at the cell membrane. Using genetic approaches, they demonstrate that loss of Slitrk2 leads to reduced numbers of excitatory synapses in cultured cortical neurons and that some mutated forms of Slitrk2 do not rescue this defect. They also examine the effect of Slitrk2 deletion in mice on a battery of behavioural assays and find that some of the defects observed, such as impaired spatial memory and abnormal gate, phenocopy some of the behaviours observed in some of the patients carrying Slitrk2 variants. In contrast, other behavioural changes observed in these mice, such as reduced anxiety, are opposite to the clinical manifestations in these patients.

This a comprehensive study describing a thorough molecular analysis of the effect novel genetic mutations identified in humans have on Slitrk2 function at synapses. Examining the effect of all the mutants on cell surface targeting, on binding to PTPdelta, and on rescuing of Slitrk2 knock-out neurons represents an impressive amount of work. However, due to the fact that the inactive Slitrk2 variants are not properly targeted to the cell membrane, there is limited new insight into Slitrk2 function that can be acquired by studying these variants in the assays used (i.e. Hemisynapse assay and cortical neuron cultures). If these mutants are not expressed at the surface of neurons, it is to be expected that they would represent loss of function mutants. This is very similar to several mutations previously identified in multiple Slitrk family members, including in Slitrk2 (as previously shown by the authors, Kang et al. 2016). The most intriguing variant is the T312A, which despite being expressed at the surface and binding PTPdelta, does not rescue synapse formation in the absence of Slitrk2. A better understanding of how this mutation affects Slitrk2 function would have hopefully provided new insight into Slitrk2 function at the synapse and increased the impact of the manuscript.

Major points :

1. The authors appear to suggest that the T312A variant and the variants that are not expressed at the membrane may have an effect on TrkB signalling based on Slitrk2 over expression experiments in

cortical neuron culture. The link with TrkB signalling is very tenuous and it is difficult to imagine how these mutants could affect TrkB signaling based on the presented data.

First, if Slitrk2 acts to repress TrkB expression/signaling under physiological circumstances, you would expect increased levels of TrkB in neurons isolated from Slitrk2 mutant mice. Is this the case?

Second, if the mutants have a dominant-negative effect on Slitrk2 regulation of TrkB expression, you would expect that expression of these mutants in wild-type cortical neurons, expressing endogenous levels of Slitrk2, would increase TrkB expression.

Third, it is unlikely that the mutants act by inhibiting the ability of Slitrk2 to control TrkB expression as all these mutants were identified in male patients (with only one copy of the X-linked Slitrk2 gene).

2. A characterization of the structure and function of synapses in the hippocampus of the Slitrk2-Nestin-Cre is needed to assess whether there is a correlation between synaptic function and the impaired spatial learning and memory observed in these mice. Is there a reduction in excitatory synapses in the hippocampus of these mice (as would be predicted based on the authors previous knock-down experiment with AAV injections (Han et al. 2019))?

Minor points :

1. In Figure 5H, a residual band with a similar size to Slitrk2 is observed in lysates from the Slitrk2 mutant. This suggests that the polyclonal antibody they generated may partly cross-react with another member of the Slitrk family.

2. A short discussion of the neuronal populations that may be affected by Slitrk2 loss, in the context of the abnormal gate, would be appreciated by the reader.

Reviewer #4 (Remarks to the Author):

This is a very thorough study demonstrating SLITRK2 variants in a neurodevelopmental disorder, with elaborate studies in vitro, in-vivo and ex-vivo in attempt to demonstrate pathogenicity of the mutations and the mechanisms through which pathogenicity is generated. The manuscript is well written and discussed. The findings are significant and novel. My review focuses on the human studies rather than the functional / mouse studies.

The manuscript asserts that a variable intellectual disability phenotype affecting 8 males in 7 families is caused by 7 different hemizygous mutations in the X-chromosome gene SLITRK2. Other, non-pathogenic hemizygous SLITRK2 variants that are found in the gnomAD database serve as “negative controls” in some of the functional studies.

While the studies are extremely thorough and extensive, a few concerns remain:

1. Phenotypic variability: All patients exhibited intellectual disability and behavior problems. However, there is microcephaly vs. macrocephaly in different patients; spasticity / spastic diplegia / dystonic diplegia in 3 patients yet not in 5 others; the intellectual disability ranges from borderline / mild to severe in different patients; there is some variability in the MRI findings (though not drastic). It should be noted, though, that phenotypic variability – even at times more extensive than here, is often reported for mutations in the same gene and even for precisely the same mutation. This should be reflected at least in a short sentence in the discussion.

2. The 7 heterozygous variants presented as pathogenic have been obtained through GeneMatcher – from a pool of thousands of intellectual disability cases. Thus, it is crucial to prove that they are pathogenic, especially in light of the variable phenotype and as there are non-pathogenic heterozygous variants of this gene in gnomAD. The authors demonstrate in-silico that the 7 variants are predicted (based on various software as well as based on the solved SLITRK1 structure) to be functional. Moreover, they show through elaborate functional studies in-vitro and ex-vivo, that the different variants affect various functions of SLITRK2: surface trafficking / dendritic targeting / rescue of excitatory synapse development and transmission in Slitrk2-cKO hippocampal neurons / ligand binding / secretion / synaptogenic activity in cultured neurons of the encoded protein. However, unless I missed significant data, functional effects are shown for 5 of the 7 variants: L74S, P374R, R426C, E461* and T312A, but not for E210K and V511M. This does appear in the discussion, mentioning that various software predict these two variants to be deleterious and that perhaps their pathogenicity is exerted through other mechanisms not studied – such as functions in astrocytes, where SLITRK2 is also expressed. A sentence in the discussion carefully mentioning also the possibility that these variants are perhaps not pathogenic is in place.

In summary, this is an extremely thorough beautiful study, implementing an extensive array of technologies. However, with the phenotypic variability presented, the existence in gnomAD of non-pathogenic hemizygous variants, combined with lack of functional proof for the E210K and V511M variants, careful wording is in place, especially concerning these two variants.

Minor comments:

1. In ClinVar there is a single case of duplication of SLITRK2 in intellectual disability – needs short mention: <https://www.ncbi.nlm.nih.gov/clinvar/variation/626291/>

2. Table 2 is only partly illegible (both Word and PDF files) – needs reformatting
3. Line 425 – Missens should be missense

Authors' rebuttal letter for Chehadeh *et al.*, "SLITRK2 variants associated with neurodevelopmental disorders impair excitatory synaptic function and cognition in mice", and description of changes made to the revised manuscript

We appreciate the reviewers' time and effort in evaluating our manuscript. The detailed comments have identified weaknesses in previous arguments and insufficient experimental explanations, underscoring the need for additional evidence. To respond as thoroughly as possible to reviewer criticisms, we conducted a series of additional experiments and made changes to the manuscript's presentation. We have expanded the supplemental data section to accommodate these new findings (now 9 main and 20 Supplementary figures!) and repositioned these supplemental figures to maintain their logical relationship with the text and major figures. Below are our responses to specific reviewer criticisms, with the entire reviewers' remarks shown in *italic* typeface and our responses and descriptions of the modifications made in the text in **bold blue** typeface.

Reviewer Comments:

Reviewer 1: *This is the Functional characterization of several patient mutations in SLITRK2 identified by human genetics studies. Cellular and molecular characterization combined with behavioral and electrophysiological studies show differences among how different patient mutations could affect differently brain function make this study highly relevant and broad significance to the fields of neurodevelopmental disorders, neurotrophin, and human genetics field. While the role of SLITRKs in synaptogenesis has been reported before this work is really the first that does such an extensive study in terms of carefully characterizing the effect of different disease variants therefore it has broad appeal. The work supports the authors claims and conclusions at large, but there are some specific points that I point out below that need to be revised and further supported as I think there are some experiments that require revisions and need to be expanded to solidify their conclusions. Sufficient detail is provided for the different experiments presented*

We very much appreciate the reviewer's overall positive assessment of our manuscript and appraisal of our study. We hope that the additional data acquired during the revision process completely address the reviewer's remaining concerns.

Specific comments per figure:

Figure 1 - can the authors show sequencing traces showing the mutations maybe put that in the supplementary material?

We present sequencing traces showing the specific mutations in Figure S1 of the revised manuscript. In the course of our in-depth review of Sanger sequencing traces, we noticed that the T312A variant was inherited from the mother (i.e., not *de novo*), contrary to the initial report from clinicians. We apologize for this unintended error, which has been corrected in the revised manuscript, although we would note that presence of this variant in a heterozygous state in the mother (as L74S) does not disprove its involvement in the neurodevelopmental disorder (as *SLITRK2* is located on the X-chromosome).

Figure S1

Figure 2 - minor point make the quadrants bigger are all at the same scale?
We have made the suggested changes in Figure 2 of the revised manuscript.

Figure 2

Figure 3C this is a very nice experiment showing changes in surface localization have they tried staining for other membrane markers for the mutants that are changing to further confirm /validate the changes in surface localization?

In response to the reviewer's suggestion, we now present immunostaining data obtained using anti-transferrin receptor antibodies that validate the surface localization of SLITRK2 mutants associated with neuropsychiatric diseases (Figure S4 in the revised manuscript).

Figure S4

Figure 4

In heK cells the ones that are changing are L74S, P374R, p426C, E361* Those are the ones that are also changing in the neurons (WT). But have they looked at the variants that are changing and use them to try to rescue the KO neurons that would be even more telling or further proof of what they are seeing by overexpression. Is it possible to make less dense cultures or take images of neurons that are in regions where there are less neurons wondering if the lack of HA signal in the dendrites is because the arbors are reduced on those neurons. Also have they try staining for golgi, er markers on the neurons in which they express the different pathogenic variants to show the mechanism of retention in the cell bodies?

In response to the reviewer's suggestion, we performed immunostaining experiments using Slitrk2 conditional knockout (KO) cultures in conjunction with a subset of SLITRK2 pathogenic variants, reaffirming the conclusions of the original overexpression experiments (presented in Figure S9). We also measured dendritic branching in neurons expressing SLITRK2 WT or its pathogenic variants to ensure that overexpression of SLITRK2 variants did not influence the dendritic arbors in neurons (presented in Figure S8). In addition, we performed immunostaining for the *cis*-Golgi marker GM130 to examine whether pathogenic SLITRK2 variants are intracellularly trapped and found no phenotypes (presented in Figure S10).

Figure S8

Figure

Figure S9

S10

Figure 5

Its very clear that slitrk2 is pretty much gone and that there is a defect that is very specific to glua2 in terms of protein levels and while there does not seem to be a change in numbers of neurons at the hippocampus from the sections the authors show there seems to be thinning of the cortex. Have they measured this? Or weigh the brains to see if compared to body weights there is a difference?

In response to the reviewer's various questions, we quantified cortical thickness and found no difference between control and Nestin-Slitrk2 mice (presented in Figure 5i). We also weighed control and Nestin-Slitrk2 mouse brains and found no difference (presented in Figure 5f).

Since Nestin CRE will target not only the progenitors for excitatory neurons but also to inhibitory interneurons have they look to see if there are any changes in inhibitory interneurons (numbers of cells or marker protein levels?)

The reviewer raises a valid point. To address it, we quantified GAD67+ inhibitory interneurons in the hippocampal CA1 region of control and Nestin-*Slitrk2* mice, but found no differences in inhibitory interneurons between the two genotypes (presented in Figure 5j, k)

Figure 5

Figure 6

Why was this experiment done using the CRE virus on the primary cultures of the *SLITRK2* floxed mice? Why did the authors did not just harvest the neurons from the *SLITRK2* floxed/nestin CRE animals and use that directly? Wouldn't there be potential differences due to a more accurate knockout on the cultures with the experiment they use?

There was no specific reason to employ *Slitrk2*-floxed mouse cultures in the experiments presented in the original Figure 6. In response to the reviewer's suggestion, we prepared hippocampal cultures from control and Nestin-*Slitrk2* mouse pups and performed both immunostaining and electrophysiology experiments. We found that *Slitrk2* deletion consistently reduced the number of excitatory synapses and the frequency of mEPSCs without affecting the number of inhibitory synapses or frequency/amplitude of mIPSCs (presented in Figures S14 and S15). These results reinforce the previous conclusions that *Slitrk2* loss-of-function manifests specifically at excitatory synapses.

Also in panel B they quantify the density the effect is clear from looking at the images the puncta in some of the mutants seems to be larger have they measure the size of the puncta? have they measure the size of the puncta?

We had already measured puncta size but did not present these data in the original manuscript. In response to the reviewer's question, we now include these data in the revised manuscript (presented in Figure S14e).

Figure S14

Figure S15

For panel E could the authors have a supplementary table showing the statistics for all the comparisons, some of the mutants that are compare to the CRE seem to have a smaller frequency compared to the control (delta CRE) so would be good to be able to see if they are in any way different from the delta CRE

As suggested by the reviewer, we now include all information regarding statistical comparisons in the revised manuscript (Table S2).

Figure 7 their evidence for the TrkB connection needs to be expanded to and made clearer in order to solidify these data

- First they need to also look at the other 2 mutants in which they see a phenotype to see if the effect is equal across mutations

In response to the reviewer's suggestion, we examined whether overexpression of two other SLITRK2 variants (R426C and E461X) also altered TrkB protein levels in cultured neurons. These experiments reproduced the initial observations, namely that overexpression of SLITRK2 WT significantly decreased the level of TrkB full-length protein (TrkB-FL). We further found that these additionally tested SLITRK2 variants behaved similarly to L74S and T312A variants, abrogating the negative regulation of TrkB levels by SLITRK2 WT. During the revision process, we also analyzed the levels of truncated TrkB protein (TrkB-T), and found that overexpression of SLITRK2 WT, but not its pathogenic variants, markedly increased its levels, implying that increased SLITRK2 might trigger degradation and/or cleavage of TrkB. These new results are presented in Fig. 7a, b of the revised manuscript.

Figure 7

- Second they need to look at phosphorylation of trkb as a ratio of total trkb from the blot it seems they are only looking at total trkb

In response to this suggestion, we performed additional immunoblotting experiments using anti-pTrkB antibodies that specifically detect Y515-phosphorylated TrkB. Strikingly, no immunoreactive bands were detectable, likely because activated TrkB protein levels are extremely low under basal conditions (presented in Fig. 7a, c of the revised manuscript).

• Third, Since there are other isoforms of TrkB like TrkB -SHC they have to clarify why they are not looking into these other isoforms

This is an important point. Please note that TrkB-T and the TrkB isoform lacking a tyrosine kinase catalytic domain cannot be discriminated because the molecular sizes of these two protein species are very similar and because the anti-TrkB monoclonal antibody was produced by immunization with a synthetic peptide surrounding Pro50 of the human TrkB protein.

Figure 8 - interesting behavioral data could the authors specify if the studies were done only in males since the condition is X-linked?

As indicated in the Methods section of the original manuscript, all behavioral experiments were performed using male mice.

Figure S3, would be good to have an internal control for lysates and if possible for secreted proteins while the effect is clear it would be good to have the values normalized to internal protein control and then that number normalized to the WT levels. Also did they discuss why is E461* so different compared to the other 2 mutants?

Figure S5

We used β -actin as an internal control for lysates (Fig. S5b of the revised manuscript). However, because it cannot be used as an internal control for media, we repeated these experiments by co-expressing an HA-tagged neuroligin-2 (Nlgn2) construct in HEK293T cells. Immunoblotting with anti-HA antibodies revealed comparable expression levels of SLITRK2 WT and the tested variants in both lysates and media, indicating equal loading of proteins (Fig. S5b of the revised manuscript). With regard to the SLITRK2 E461X variant, we repeated the same experiments with newly prepared sets of SLITRK2 plasmids and found that the secretion level of the SLITRK2 E461X variant was comparable to that of SLITRK2 WT (presented in Fig. S5c with new quantification result).

Figure S5, this is a very interesting finding using the heterologous system can the authors analyze inhibitory synapses in the KO animals? Maybe puncta of GAD67?

We appreciate the reviewer's appraisal of our data presented in Figure S5. As suggested by the reviewer, we examined whether SLITRK2 deletions affected inhibitory synapse properties (i.e., numbers and basal synaptic transmission), but found no alterations (presented in Figure S14, S15 and S19).

Figure S19

Is the lack of dendritic localization of SLITRK pathogenic variants due to

Unfortunately, we were unable to address this comment because it was truncated; hopefully it was a minor issue or one that was addressed in our response to another comment.

Overall this is a very exciting study that includes human genetics, mouse behavior and cellular characterization on the function of *SLITRK2* disease variants.

Again, we very much appreciate the reviewer's positive comments and constructive suggestions for improving our original manuscript.

Reviewer 2: In this manuscript, Chehadeh et al. report a number of new missense variants that further implicate the transmembrane synaptic adhesion protein *Slitrk2* in the pathogenesis of X-linked intellectual disability. The authors examine the biochemical interactions, cell surface transport, and synapse forming activities of these missense mutants using mainly in vitro approaches, and they also perform behavioral analysis of newly generated conditional *Slitrk2* knock out mice. The paper is ambitious, technically rigorous, and contributes significantly to our understanding of the genetic basis of X-linked intellectual disability (XLID) and related neurodevelopmental disorders. It also reports the production of two valuable research reagents in the synaptic adhesion research community for which no commercially reliable products exist – a *Slitrk2* antibody, as well as a *Slitrk2* conditional knock out mouse. The manuscript is remarkably well written – both comprehensive and easy to follow. I have only three moderate concerns and a few minor suggestions to communicate.

We very much appreciate the reviewer's overall positive assessment of our paper and hope that our extensive revisions will assuage any remaining concerns.

Major points:

1. In Figure 3, the authors claim that three missense variants (L74S, P374S, and R426C) do not exhibit the cell surface expression that is characteristic of WT *Slitrk2* and other missense variants. However, upon examination of the images in Figure 3C, it is not entirely convincing that the staining patterns observed for these mutants are intracellular. A similar concern is applicable to Supplementary Figure 5, where, again, the staining patterns of these mutants appears more surface than intracellular. Because this claim of a surface transport defect is an important part of the paper's premise, additional surface vs. intracellular data are warranted to strengthen this claim.

In response to the reviewer's comments, we performed a comparative analysis of surface expression of *SLITRK2* variants using surface biotinylation assays. In line with the conclusions shown in original Figure 3, we found that these three *SLITRK2* missense variants exhibited less surface expression compared with *SLITRK2* WT (presented in Figure S6).

Figure S6

2. In Figure 6, as well as Supplementary Figure 5, it is curious that the authors only report data on excitatory synapse formation, while omitting any analysis of inhibitory synapse formation. There is some suggestion in previous literature (see Takahashi et al. 2012, Figure 1) that *Slitrk2* can induce the formation of both excitatory and inhibitory synapses in a co-culture assay. Given the uncertainty regarding *Slitrk2*'s ability to induce inhibitory synapses, the inclusion of inhibitory synapse markers in these experiments would be informative.

We appreciate the reviewer's comments on Figures S5 and S6, which were echoed by comments of

Reviewer #1. Indeed, Slitrk2 induces presynaptic assembly at both excitatory and inhibitory synapses, as documented in many prior reports (Takahashi et al., 2012 *Nat Neurosci*; Yim et al., 2013 *Proc Natl Acad Sci USA*; Kang et al., 2016 *Front Mol Neurosci*). However, when analyzed in cultured hippocampal neurons, including hippocampal CA1 pyramidal neurons, Slitrk2 is specifically active at excitatory, but not inhibitory, synapses (Yim et al., 2013 *Proc Natl Acad Sci USA*; Kang et al., 2016 *Front Mol Neurosci*; Han et al., 2019 *Sci Rep*). Nevertheless, we performed an extensive series of experiments to analyze inhibitory synapse numbers and inhibitory synaptic transmission in cultured SLITRK2-KO neurons and confirmed no noticeable phenotypes at inhibitory synapses (presented in Figures S14 and S15).

3. In the discussion, the authors state that they have found an association of Slitrks 2, 3, and 5 with TrkB, but no more information is given, and no data is shown. Given that the TrkB interaction forms the basis of Figure 7, the inclusion of these data would be a valuable addition to Figure 7, or at least as a figure in the supplementary materials.

We agree with the reviewer's suggestion and now included these data in Figure S20 of the revised manuscript.

Figure S20

Minor points:

In Figure 3B, it appears at least by banding pattern that some of the mutants have altered glycosylation or other posttranslational differences. Has this been examined experimentally?

Our previous study demonstrated that the upper band in Western blots of brain samples corresponds to the mature, fully glycosylated SLITRK2 protein (Yim et al., 2013 *Proc Natl Acad Sci USA*). Glycosylation of the SLITRK2 variants, L74S, P347R and R426C, appears to be inhibited, consistent with their impaired surface trafficking behaviors (Fig. 3d in the revised manuscript). To clearly address the reviewer's question, we quantified glycosylated SLITRK2, calculated as the ratio of the upper band intensity to total band intensity (now presented in Fig. S3 of the revised manuscript).

Figure S3

In Figure 4, it is interesting that dendritic trafficking of each the four featured mutants is compromised, yet the relative amount of protein in the somatic compartment is not elevated. If expression levels of these mutants of interest are unchanged (Fig 3A) in comparison to other forms of Slitrk2, then it would seem that somatic measurements of these mutants should be elevated.

We appreciate the reviewer's close inspection of our original Figure 4. We suspected that quantification of HA immunoreactivity in the somatic compartment of the transfected neurons might be misleading, owing to signal saturation caused by overexpression. We thus repeated the same experiments, but captured the images using different confocal microscopy settings that avoid saturation artifacts. We found that somatic expression levels of SLITRK2 L74S and P374R were significantly increased relative to those of SLITRK2 WT (presented in Fig. S8a, b). In contrast, the expression level of SLITRK2 R426C was not increased in somatic compartments. One explanation for these puzzling results is that the dendritic targeting defect of SLITRK2 R426C is less severe than that of the other two SLITRK2 variants (L74S and P374R) (see Fig. 4c) and its total expression level is comparable to that of SLITRK2 WT (see Fig. 3b).

With regard to the mention of PDZ-containing scaffolds in Line 652 of the discussion, Loomis et al. 2020 should also be included as a reference.

As suggested, we have cited the Loomis et al. paper in line 652 of the revised manuscript.

All figures containing immunocytochemistry should be reformatted so as to avoid the red/green color scheme.

In response to the reviewer's suggestion, we changed the color scheme from red/green to magenta/cyan/yellow in the revised manuscript.

Reviewer 3: *In this manuscript, the authors identify novel mutations in the SLITRK2 gene associated with X-linked neurodevelopmental disorders. They demonstrate that several of these mutations lead to a loss in Slitrk2 function at synapses due to lack of receptor expression at the cell membrane. Using genetic approaches, they demonstrate that loss of Slitrk2 leads to reduced numbers of excitatory synapses in cultured cortical neurons and that some mutated forms of Slitrk2 do not rescue this defect. They also examine the effect of Slitrk2 deletion in mice on a battery of behavioural assays and find that some of the defects observed, such as impaired spatial memory and abnormal gate, phenocopy some of the behaviours observed in some of the patients carrying Slitrk2 variants. In contrast, other behavioural changes observed in these mice, such as reduced anxiety, are opposite to the clinical manifestations in these patients. This a comprehensive study describing a thorough molecular analysis of the effect novel genetic mutations identified in humans have on Slitrk2 function at synapses. Examining the effect of all the mutants on cell surface targeting, on binding to PTPdelta, and on rescuing of Slitrk2 knock-out neurons represents an impressive amount of work. However, due to the fact that the inactive Slitrk2 variants are not properly targeted to the cell membrane, there is limited new insight into Slitrk2 function that can be acquired by studying these variants in the assays used (i.e. Hemisynapse assay and cortical neuron cultures). If these mutants are not expressed at the surface of neurons, it is to be expected that they would represent loss of function mutants. This is very similar to several mutations previously identified in multiple Slitrk family members, including in Slitrk2 (as previously shown by the authors, Kang et al. 2016). The most intriguing variant is the T312A, which despite being expressed at the surface and binding PTPdelta, does not rescue synapse formation in the absence of Slitrk2. A better understanding of how this mutation affects Slitrk2 function would have hopefully provided new insight into Slitrk2 function at the synapse and increased the impact of the manuscript.*

We appreciate the reviewer's overall positive assessment of our original presentation. We also welcome the reviewer's critical comments, particularly those regarding the T312A mutant. Although the current study is, by nature, relatively descriptive in its account of the mechanistic action of SLITRK2 mutants, we would urge the perspective that the issues raised by the reviewer do not detract from the merits of the current study, which demonstrated the loss-of-function properties of SLITRK2 pathogenic variants and showed that SLITRK2 is functionally linked to TrkB in neurons. We also fully agree with the sentiment expressed in closing sentence and hope that our extensive experiments during revision assuages the reviewer's remaining concerns.

Major points :

1. *The authors appear to suggest that the T312A variant and the variants that are not expressed at the membrane may have an effect on TrkB signalling based on Slitrk2 over expression experiments in cortical neuron culture. The link with TrkB signalling is very tenuous and it is difficult to imagine how these mutants could affect TrkB signaling based on the presented data. First, if Slitrk2 acts to repress TrkB expression/signaling under physiological circumstances, you would expect increased levels of TrkB in neurons isolated from Slitrk2 mutant mice. Is this the case?*

To address the reviewer's question, we performed TrkB immunoblotting experiments using lysates of the hippocampal CA1 region from Slitrk2-cKO mice. As the reviewer surmised, we found that loss of Slitrk2 significantly increased TrkB-FL levels (presented in Figure 7c, d of the revised manuscript).

Second, if the mutants have a dominant-negative effect on Slitrk2 regulation of TrkB expression, you would expect that expression of these mutants in wild-type cortical neurons, expressing endogenous levels of Slitrk2, would increase TrkB expression.

Our interpretation is that SLITRK2 is required for maintaining an appropriate balance of TrkB protein levels and that SLITRK2 pathogenic variants do not act in a dominant-negative manner. To wit: overexpression of SLITRK2 WT increases excitatory synaptic transmission in neurons, leading to downregulation of TrkB-FL, likely accompanied by upregulation of TrkB-T through activation of calpain proteases, possibly serving to dampen heightened network activity triggered by overexpression of SLITRK2 WT. Under this scenario, overexpression of SLITRK2 pathogenic variants failed to downregulate TrkB-FL, consistent with their absence of an effect on excitatory synapse density and transmission (Fig. 6). Because the binding strength of SLITRK2 T312A to TrkB is comparable to that of SLITRK2 WT (Fig. S17a, b), SLITRK2 might function in negatively regulating TrkB levels through mechanisms apart from its direct interaction with TrkB. Future studies should dissect the nature of the crosstalk between SLITRK2 and TrkB.

Third, it is unlikely that the mutants act by inhibiting the ability of Slitrk2 to control TrkB expression as all these mutants were identified in male patients (with only one copy of the X-linked Slitrk2 gene).

Our inference is that this comment is based on the interpretation that SLITRK2 pathogenic variants act in a dominant-negative manner to inhibit the activity of endogenous SLITRK2 function. As mentioned in our response to the reviewer's previous comment, we interpret the results of our culture experiments as indicating that regulation of TrkB activity is compromised by SLITRK2 pathogenic mutations and might give rise to abnormally altered network activities and neural circuit functions. This interpretation also aligns well with our observation that SLITRK2 cKO increases TrkB-FL levels (Fig. 7c, d of the revised manuscript). Of course, as the reviewer points out, the pathogenicity of SLITRK2 variants in male patients results from elimination of SLITRK2 function.

2. *A characterization of the structure and function of synapses in the hippocampus of the Slitrk2-Nestin-Cre is needed to assess whether there is a correlation between synaptic function and the impaired spatial learning and memory observed in these mice. Is there a reduction in excitatory synapses in the hippocampus of these mice (as would be predicted based on the authors previous knock-down experiment with AAV injections (Han et al. 2019))?*

As suggested by the reviewer, it is critical to correlate Slitrk2-mediated synaptic functions and behavioral phenotypes. To address this, we stereotactically injected AAVs expressing Cre recombinase into the hippocampal CA1 and performed semi-quantitative immunohistochemistry, electrophysiological recordings, and a subset of key behavioral experiments. We found that CA1-specific Slitrk2 deletion recapitulated the phenotypes observed in cultured neurons from Slitrk2-cKO mice (presented in Figure 9c–g). Strikingly, we found that CA1-specific Slitrk2-KO mice exhibited impaired spatial memory, as assessed by the Barnes maze test (presented in Figure 9h–

l), arguing that excitatory synapse dysfunctions induced by loss of Slitrk2 might underlie impaired spatial memory in mice.

Figure 9

Minor points :

1. In Figure 5H, a residual band with a similar size to Slitrk2 is observed in lysates from the Slitrk2 mutant. This suggests that the polyclonal antibody they generated may partly cross-react with another member of the Slitrk family.

Before submitting the original manuscript, we had already characterized our in-house Slitrk2 antibody, JK177, in immunoblotting experiments, showing that it is specifically active in cell lysates expressing SLITRK2, but not those expressing other SLITRK paralogs (presented in Figure S12). To avoid unnecessary confusion, we have replaced the representative immunoblot image with a new one.

Figure S12

2. A short discussion of the neuronal populations that may be affected by Slitrk2 loss, in the context of the abnormal gate, would be appreciated by the reader.

We appreciate the reviewer's excellent suggestions. In response, we have added the following

additional text in the Discussion section of the revised manuscript highlighting the significance of revealing the neural circuits involved in motor coordination: *In general, the cerebellum is critical for coordinating movements, and its dysfunction often leads to uncoordinated walking or gait disturbance⁴⁰, a feature of patients harboring a subset of SLITRK2 variants. Although the current study primarily correlated SLITRK2 synaptic functions in the hippocampal CA1 with spatial memory performance, future studies are warranted to identify the neuronal populations and/or neural circuits embedded in other brain areas that might underpin abnormal footprint patterns seen in Nestin-Slitrk2 mice.*

Reviewer 4: *This is a very thorough study demonstrating SLITRK2 variants in a neurodevelopmental disorder, with elaborate studies in vitro, in-vivo and ex-vivo in attempt to demonstrate pathogenicity of the mutations and the mechanisms through which pathogenicity is generated. The manuscript is well written and discussed. The findings are significant and novel. My review focuses on the human studies rather than the functional / mouse studies. The manuscript asserts that a variable intellectual disability phenotype affecting 8 males in 7 families is caused by 7 different hemizygous mutations in the X-chromosome gene SLITRK2. Other, non-pathogenic hemizygous SLITRK2 variants that are found in the gnomAD database serve as “negative controls” in some of the functional studies. While the studies are extremely thorough and extensive, a few concerns remain:*

We very much appreciate the reviewer’s overall positive evaluation of our study and assessment as being “significant” and “novel”. We hope that changes made in the revised manuscript fully address the reviewer’s remaining concerns.

1. Phenotypic variability: All patients exhibited intellectual disability and behavior problems. However, there is microcephaly vs. macrocephaly in different patients; spasticity / spastic diplegia / dystonic diplegia in 3 patients yet not in 5 others; the intellectual disability ranges from borderline / mild to severe in different patients; there is some variability in the MRI findings (though not drastic). It should be noted, though, that phenotypic variability – even at times more extensive than here, is often reported for mutations in the same gene and even for precisely the same mutation. This should be reflected at least in a short sentence in the discussion.

The individuals we described with rare SLITRK2 variants exhibited neurodevelopmental disorders (NDD), including recurrent clinical features such as speech delay, intellectual disability (ID), and behavioral problems. Some common brain MRI abnormalities were also observed (e.g., ventricular dilation, white matter volume loss), although these were not specific and were frequently observed in ID patients. However, the affected individuals showed some phenotypic variability, as reflected in ID severity ranging from borderline to severe; moreover, only 3/8 patients had seizures, 3/8 had dystonia or spasticity, and half had unsteady gait. In addition, occipitofrontal circumference was highly variable, with two patients exhibiting macrocephaly and one patient exhibiting microcephaly (presented in Table 1). These findings are not surprising since phenotypic variability is often reported in rare genetic conditions associated with ID (see Baer et al. *Clinical Genetics* 2018). This clinical heterogeneity can be observed among patients carrying pathogenic variants in the same gene and even those with exactly the same variant, for instance in the same family (Piton et al, *Hum Mol Genet* 2008). Thus, our interpretation is that other genetic and/or environmental modifying factors are also involved. This is conveyed in the following sentence, added to the Discussion section of the revised manuscript: “Such clinical heterogeneity among individuals is often reported in NDD for mutations in the same gene and even for precisely the same mutation^{37,38}, suggesting that other genetic and/or environmental modifying factors might also be involved”.

2. The 7 heterozygous variants presented as pathogenic have been obtained through GeneMatcher – from a pool of thousands of intellectual disability cases. Thus, it is crucial to prove that they are pathogenic, especially in light of the variable phenotype and as there are

non-pathogenic heterozygous variants of this gene in gnomAD. The authors demonstrate in-silico that the 7 variants are predicted (based on various software as well as based on the solved SLITRK1 structure) to be functional. Moreover, they show through elaborate functional studies in-vitro and ex-vivo, that the different variants affect various functions of SLITKR2: surface trafficking / dendritic targeting / rescue of excitatory synapse development and transmission in Slitrk2-cKO hippocampal neurons / ligand binding / secretion / synaptogenic activity in cultured neurons of the encoded protein. However, unless I missed significant data, functional effects are shown for 5 of the 7 variants: L74S, P374R, R426C, E461 and T312A, but not for E210K and V511M. This does appear in the discussion, mentioning that various software predict these two variants to be deleterious and that perhaps their pathogenicity is exerted through other mechanisms not studied – such as functions in astrocytes, where SLITRK2 is also expressed. A sentence in the discussion carefully mentioning also the possibility that these variants are perhaps not pathogenic is in place. In summary, this is an extremely thorough beautiful study, implementing an extensive array of technologies. However, with the phenotypic variability presented, the existence in gnomAD of non-pathogenic hemizygous variants, combined with lack of functional proof for the E210K and V511M variants, careful wording is in place, especially concerning these two variants.*

We completely agree with the reviewer. We currently have no evidence that E210K and V511M mutations affect SLITRK2 function and cannot claim at this point that they are pathogenic. That is why they are classified as “variant of unknown significance” in Table 2; in this table, only L74S, P374R, R426C, E461* and T312A are indicated as likely pathogenic according to ACMG criteria. To make this point clearer, we have slightly modified a sentence in the original Discussion section to read as follows: “Because our various analyses did not reveal any prominent phenotypes in the case of E210K and V511M (predicted to be deleterious by in silico programs), their pathogenicity remains uncertain, although it is still possible that these variants are involved in SLITRK2-associated pathophysiology via as-yet unidentified mechanism(s) encompassing early brain developmental processes”. Nevertheless, we decided to retain clinical information for individuals with E210K and V511M variants in Table 1, which presents clinical similarities with other individuals carrying likely pathogenic variants.

Minor comments:

1. In ClinVar there is a single case of duplication of SLITRK2 in intellectual disability – needs short mention: <https://www.ncbi.nlm.nih.gov/clinvar/variation/626291/>

This link refers to a large duplication of 138 Mb encompassing an important portion of the X chromosome. We were not able to find any duplication limited to SLITRK2 in ClinVar. We have added a sentence to address this point in the revised manuscript.

2. Table 2 is only partly illegible (both Word and PDF files) – needs reformatting

As suggested, we have reformatted Table 2 to make it legible in the revised manuscript.

3. Line 425 – Missens should be missense

We have corrected this error in the revised manuscript.

Again, we thank the reviewers for their careful assessment of our paper and hope that the revised manuscript is now acceptable for publication in *Nature Communications*.

REVIEWERS' COMMENTS

Reviewer #1 (Remarks to the Author):

The revised manuscript on SLITRK2 has extensively improved the original work and provides additional novel mechanistic insights into how different pathogenic variants in SLITRK2 could be affecting neuronal development by modulating two different TrkB isoforms and how they are distinctively affecting excitatory vs. inhibitory neurons.

The authors have used a careful methodological approach to address all of this reviewer's comments/concerns, and in the process of addressing all of the other reviewers comments they have generated an outstanding and rigorous body of work that contributes extensively to our understanding on how SLITRK2 functions during broader neuronal development and how its dysfunction leads to disease.

Reviewer #2 (Remarks to the Author):

The authors did an excellent job of addressing the suggestions and concerns brought forth in the previous reviews. The revised manuscript is ambitious, technically rigorous, and impressive in its scope. This work contributes significantly to our understanding of the genetic basis of X-linked intellectual disability (XLID) and related neurodevelopmental disorders. I have no further concerns to relay to the authors.

Reviewer #3 (Remarks to the Author):

The authors have addressed all my technical/data comments with further discussion and new experiments. I am satisfied with their responses.

Reviewer #4 (Remarks to the Author):

The comments have been addressed.

Still, slightly more cautious wording in the Discussion would be appreciated in three points:

1. Abstract:

In place of:

Strikingly, these mutations abolished the ability...

Suggested wording:

Strikingly, these variants abolished the ability...

2. Line 432:

In place of:

In the current study, we identified SLITRK2 variants as a cause of NDD characterized by...

Suggested wording:

In the current study, we identified SLITRK2 variants in NDD characterized by...

3. Line 486:

In place of:

In conclusion, we demonstrated that nonsense and missense variants in SLITRK2 lead to a novel form of X-linked NDD characterized.....

Suggested wording:

In conclusion, we demonstrated nonsense and missense variants in SLITRK2 in a novel form of X-linked NDD characterized.....

Authors' rebuttal letter for Chehadeh *et al.*, "SLITRK2 variants associated with neurodevelopmental disorders impair excitatory synaptic function and cognition in mice", and description of changes made to the revised manuscript

We are grateful to the reviewers for the careful evaluation and critical analysis of this manuscript. Our responses to specific comments of the reviewer are provided below, with the complete reviewers' comments shown in *italic* typeface and the responses and descriptions of the changes made in the manuscript shown in **bold blue** typeface.

Reviewer Comments:

Reviewer 1: *The revised manuscript on SLITRK2 has extensively improved the original work and provides additional novel mechanistic insights into how different pathogenic variants in SLITRK2 could be affecting neuronal development by modulating two different TrkB isoforms and how they are distinctively affecting excitatory vs. inhibitory neurons. The authors have used a careful methodological approach to address all of this reviewer's comments/concerns, and in the process of addressing all of the other reviewers comments they have generated an outstanding and rigorous body of work that contributes extensively to our understanding on how SLITRK2 functions during broader neuronal development and how its dysfunction leads to disease.*

We thank the reviewer for support for publication of our revised manuscript.

Reviewer 2: *The authors did an excellent job of addressing the suggestions and concerns brought forth in the previous reviews. The revised manuscript is ambitious, technically rigorous, and impressive in its scope. This work contributes significantly to our understanding of the genetic basis of X-linked intellectual disability (XLID) and related neurodevelopmental disorders. I have no further concerns to relay to the authors.*

We thank the reviewer for support for publication of our revised manuscript.

Reviewer 3: *The authors have addressed all my technical/data comments with further discussion and new experiments. I am satisfied with their responses.*

We thank the reviewer for support for publication of our revised manuscript.

Reviewer 4: *The comments have been addressed. Still, slightly more cautious wording in the Discussion would be appreciated in three points:*

1. Abstract:

In place of:

Strikingly, these mutations abolished the ability....

Suggested wording:

Strikingly, these variants abolished the ability....

2. Line 432:

In place of:

In the current study, we identified SLITRK2 variants as a cause of NDD characterized by...

Suggested wording:

In the current study, we identified SLITRK2 variants in NDD characterized by...

3. Line 486:

In place of:

In conclusion, we demonstrated that nonsense and missense variants in SLITRK2 lead to a novel form of X-linked NDD characterized....

Suggested wording:

In conclusion, we demonstrated nonsense and missense variants in SLITRK2 in a novel form of X-linked NDD characterized.....

To address the reviewer's suggestions, we reworded the indicated three parts as instructed in the revised manuscript.